# Blobby is a synaptic active zone assembly protein required for memory in *Drosophila*

J. Lützkendorf [1,12], T. Matkovic-Rachid[1,12], S. Liu[2], T. Götz[1], L. Gao[1], O. Turrel [1], M. Maglione [1,3], M. Grieger[1], S. Putignano[1], N. Ramesh [1], T. Ghelani [1,4], A. Neumann[1], N. Gimber [5], J. Schmoranzer[5], A. Stawrakakis[1], B. Brence[6], D. Baum [6], Kai Ludwig [7], M. Heine[8], T. Mielke [9], F. Liu [4], A. M. Walter [4,10], M. C. Wahl [2] & S. J. Sigrist [1,11] ✉

At presynaptic active zones (AZs), scaffold proteins are critical for coordinating synaptic vesicle release and forming essential nanoarchitectures. However, regulatory principles steering AZ scaffold assembly, function, and plasticity remain insufficiently understood. We here identify an additional *Drosophila* AZ protein, "Blobby", essential for proper AZ nano-organization. Blobby biochemically associates with the ELKS family AZ scaffold protein Bruchpilot (BRP) and integrates into newly forming AZs. Loss of Blobby results in fewer AZs forming, ectopic AZ scaffold protein accumulations ("blobs") and disrupts nanoscale architecture of the BRP-AZ scaffold. Functionally, *blobby* mutants show diminished evoked synaptic currents due to reduced synaptic vesicle release probability and fewer functional release sites. Blobby is also present in adult brain synapses, and post-developmental knockdown of Blobby in the mushroom body impairs olfactory aversive memory consolidation. Thus, our analysis identifies an additional layer of AZ regulation critical for developmental AZ assembly but also for AZ-mediated plasticity controlling behavior.

Active zones (AZs) are cellular platforms that govern presynaptic release function. These nanoscale-patterned macromolecular architectures serve as major signaling hubs for synaptic information transfer and storage[1–4]. At AZ membranes, synaptic vesicles (SVs) are released and replenished at high speeds, with subtle deficits in release and/or replenishment rates provoking a detrimental impact on the behavioral and thus organismal level. In ultrastructural and molecular terms, AZs are decorated by electron dense "scaffolds", which form from extended cytoplasmic proteins belonging to only a few evolutionary conserved families[4]. These scaffold structures help the recruitment of new SVs to AZ membrane[5–7] where they get biochemically primed for release at sites enriched for (m)Unc13 family release factor proteins which operate in conjunction with nanoscopic clusters of conserved voltage-gated Ca²⁺ channels[8–11].

To execute their essential biological functions, AZ membranes must be patterned at the nanoscale level, which means in the range of a few tens of nanometers. Genetic analyses in model organisms such as *C. elegans*, *Drosophila*, and mice have shed light on the critical role of

¹Freie Universität Berlin, Institute for Biology and Genetics, Berlin, Germany. ²Freie Universität Berlin, Institute of Chemistry and Biochemistry/Structural Biochemistry, Berlin, Germany. ³Freie Universität Berlin, Institute for Chemistry and Biochemistry, SupraFAB, Berlin, Germany. ⁴Leibniz-Forschungsinstitut für Molekulare Pharmakologie, Campus Berlin-Buch, Berlin, Germany. ⁵Charité- Universitätsmedizin, Advanced Medical Bioimaging Core Facility, Berlin, Germany. ⁶Zuse Institute Berlin, Department of Visual and Data-Centric Computing, Berlin, Germany. ⁷Freie Universität Berlin, Institut für Chemie und Biochemie, Forschungszentrum für Elektronenmikroskopie, Berlin, Germany. ⁸Institute of Developmental Biology and Neurobiology, Johannes Gutenberg University Mainz, Mainz, Germany. ⁹Max Planck Institute for Molecular Genetics, Berlin, Microscopy and Cryo-Electron Microscopy Service Group, Berlin, Germany. ¹⁰University of Copenhagen, Department of Neuroscience, Copenhagen, Denmark. ¹¹Charité Universitätsmedizin, NeuroCure Cluster of Excellence, Charitéplatz, Berlin, Germany. ¹²These authors contributed equally: J. Lützkendorf, T. Matkovic-Rachid. ✉e-mail: stephan.sigrist@fu-berlin.de

conserved scaffold proteins, notably the ELKS (BRP in *Drosophila*) and Liprin-α families, in driving the AZ assembly process[12–20]. Previous genetic and intravital imaging analyses have established functional and temporal sequences of AZ scaffold protein accumulation[12,21–23]. However, despite their fundamental importance for circuit development and behaviorally relevant plasticity, the precise mechanisms by which functional AZ scaffolds assemble and remodel in vivo remain insufficiently understood. Along these lines, proteins evolved to specifically regulate the assembly and remodeling of AZ scaffolds have yet to be identified.

We here introduce "Blobby," an AZ scaffold localized protein in *Drosophila*. Comprising coiled-coil and intrinsically disordered domains, Blobby integrates into newly assembling AZs by associating with the ELKS family protein Bruchpilot (BRP). Loss of Blobby undermines the nanoscale architecture of the BRP-based AZ scaffold, and results in aberrant protein accumulations ("blobs"), as well as reduced SV release. Blobby knockdown within the intrinsic neurons of the mushroom body impairs olfactory memory consolidation, highlighting its role in both developmental AZ assembly and presynaptic plasticity important for behavior.

## Results

### Biochemical identification of an additional active zone scaffold protein in *Drosophila*

At *Drosophila* AZs, the ELKS-family scaffold protein BRP plays a crucial role in forming SV release sites by promoting the nanoscale localization of Unc13A, a member of the (m)Unc13 family of proteins[8,10,16], and by clustering $Ca^{2+}$ channels at the AZ membrane via its N-terminal region[16]. Additionally, BRP's elongated shape, approximately 80 nm in length, facilitates SV targeting toward the SV release sites[24–26]. Additionally, BRP at *Drosophila* synapses fulfills the criteria of a "master AZ scaffold protein" whose levels determines the local (AZ) and global (neuron-wide) pools of Unc13A[27].

In our pursuit to identify additional proteins involved in the developmental assembly of AZ scaffolds, we conducted anti-BRP immunoprecipitations from *Drosophila* brain synaptosome preparations[28] which we subjected to proteomic analysis (Fig. 1A). As anticipated, we identified known BRP interactors including RIM-BP[18] and Unc13A[8] (Fig. 1A; Supplementary table I). However, the most enriched protein we detected was a protein encoded by the *CG42795* locus (Flybase release FB2024_04, Fig. 1A; Supplementary table I). To capture an interesting aspect of its mutant phenotype described below, we henceforth refer to the encoded protein as "Blobby".

We generated antibodies against this novel factor, to detect the protein in brain homogenate Western blots (Fig. 1B, C). *CG42795/blobby* encodes a sizable protein (several isoforms predicted by Flybase[29] in the size range of 250-300 kD), distinguished by its extensive intrinsically disordered regions (IDRs) and coiled-coil (blue triangles) domains (Fig. 1D). Additionally, it contains a Tre-2, Bub2 and Cdc16 (TBC)-type putative Rab-GAP domain near its N-terminus (Fig. 1D), for which dendrogram analysis suggests homology to the human protein TBC1D30 (Fig. S1A). Western probing in *Drosophila* brain extracts with either the ex8b Ab or the C-term Ab (for position of epitopes see Fig. S1B) detected several bands in the predicted size range (Fig. 1B). To independently validate our proteomics/mass spectrometry results, we also probed the BRP immuno-precipitates from *Drosophila* brain synaptosomes with anti-Blobby[C-term] antibody. Indeed, anti-Blobby reactivity was detected in the BRP immuno-precipitates (Fig. 1C, right lane) but not in controls using irrelevant immunoglobulins (Fig. 1C, middle lane).

To functionally study Blobby, we generated null mutants by sequentially deleting two major segments of the *blobby* open reading frame via CRISP/Cas9 editing, as indicated in the gene locus map (Fig. S1B). As expected, the Western blot signals for both anti-Blobby antibodies were eliminated in these *blobby*[Null] mutants (Fig. 1B).

Confocal microscopy analysis (Fig. 1E) of larval neuromuscular junction (NMJ) terminals, identified by horse radish peroxidase (HRP) staining, showed synaptic staining for Blobby, which was absent in *blobby*[Null] mutant larvae.

Given the Co-IP of Blobby with BRP, and its immunofluorescence distribution, we wondered whether Blobby might be an additional AZ scaffold protein. Indeed, immunostaining of Blobby with bona fide AZ scaffold proteins, BRP or RIM-BP, revealed colocalization between Blobby and these AZ scaffold components (Fig. 2A). Observing this AZ scaffold localization, we sought to genetically confirm that this Blobby AZ signal derives from presynaptic motoneuron expression. To address this, we integrated a marker cassette flanked by KDRT recombinase sites into an exon of *blobby* locus using Crispr/Cas9 *(blobby-STOP-ALFA*, Fig. S1B). As expected, Blobby and ALFA-staining were abolished in this line (Fig. 2B, upper row). However, after *ok6-Gal4* mediated KD-recombinase expression in motoneurons and subsequent precise out-recombination of the KDRT-STOP-KDRT cassette, AZ expression of ALFA-tagged Blobby was restored (*ok6>blobby-ALFA*, Fig. 2B, lower row). In contrast, the expression of KD recombinase specifically within the postsynaptic muscle cells did not result in any NMJ or muscle Blobby staining (Fig. S2). Thus, Blobby protein at presynaptic NMJ AZs evidently is derived from the motoneurons, the presynaptic partner cell of the NMJ.

### Blobby incorporates late into newly forming NMJ active zone scaffolds

To further explore the presynaptic AZ localization of Blobby, we turned to time-gated stimulated emission depletion (gSTED) microscopy[30]. We here utilized our on-locus Blobby-GFP line (labeled GFP in Fig. S1B) stained with anti-GFP along with the Blobby[ex8b] antibody and the BRP[NC82] monoclonal antibody (Fig. 2C). This triple-channel STED experiment revealed that both Blobby epitopes were entirely confined within the BRP-positive AZ scaffold (Fig. 2C), particularly noticeable in planar, *en face* imaged AZs (see magnifications labeled with I). In vertically imaged AZs (labeled with II), average distances between all three epitopes were measured and found to be about 20 nm (for exact values see Fig. 2C legend and source data). Thus, the Blobby protein integrates into the AZ-scaffold close to the BRP[NC82] epitope. Given that the BRP[NC82] epitope marks the distal end of the T-Bar AZ scaffold in a distance of about 70 nanometers from the AZ localized SV release sites[31], we conclude that Blobby is an additional AZ scaffold component colocalizing with BRP in the AZ scaffold of developing *Drosophila* NMJ synapses.

We next investigated whether the localization of Blobby within AZ scaffolds might rely on BRP. Thus, we stained *brp*[Null] mutant NMJ terminals with Blobby[ex8b] and RIM-BP[C-term] antibodies (Fig. 2D–F). RIM-BP staining was selected because it can detect AZ scaffolds even in the absence of BRP[18]. Confocal images of controls showed that Blobby colocalized with RIM-BP (Fig. 2D), as expected (Fig. 2A). However, the degree of co-localization appeared clearly reduced at *brp*[Null] mutant terminals (Fig. 2D, see magnifications to the right), evident also in quantifications using two distinct colocalization scores (Fig. 2E, F). Thus, Blobby's localization within AZ scaffolds appears to be impaired in the absence of BRP. Notably, however, in the absence of BRP the bulk of Blobby still reached a position close to the AZ scaffolds implying that its long-range transport is at least to a degree independent of BRP (also see discussion).

Intact *Drosophila* larvae allow for live imaging of synapse assembly at consecutive time points, using fluorescently labeled synaptic proteins[32–34]. At *Drosophila* larval NMJs, AZ scaffold assembly is initiated by the accumulation of Liprin-α and Syd-1 ("early scaffold proteins"), while BRP is incorporated later into newly forming AZs[22]. Given Blobby's close association with BRP, we performed intravital co-imaging of both proteins at two consecutive time points, 24 h apart,

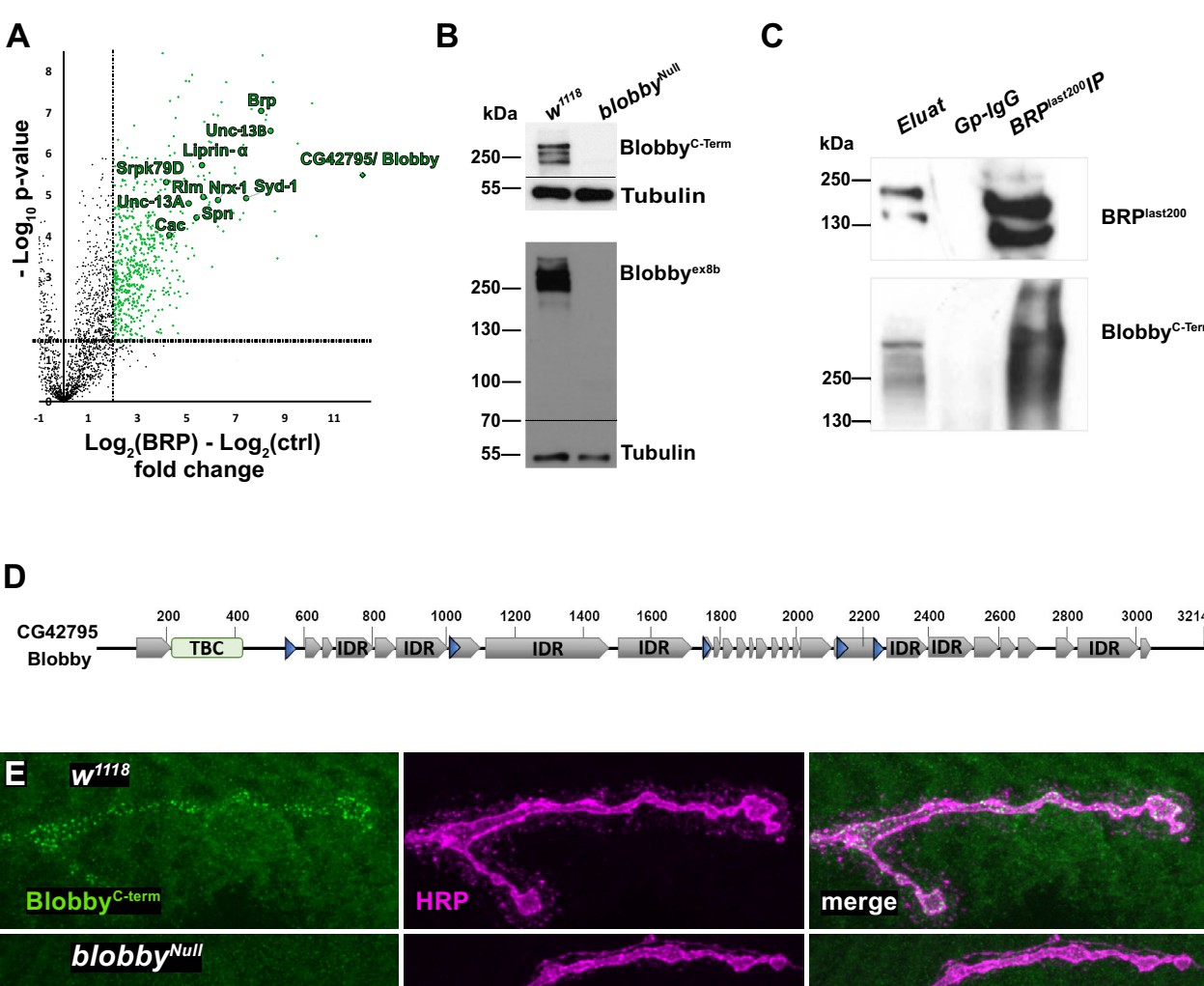

**Fig. 1 | Identification of a protein enriched in the AZ scaffold protein complex.**
**A** Volcano plot of LFQ MS data plotting the logarithmic difference from quadruplicate experiments of coprecipitated protein levels from the BRP-IP compared with the IgG control. *X*-axis represents $\log_2$ fold-change values, *y*-axis displays $-\log_{10}$ of the *p*-value, *p*-values < -$\log_{10}(0.05)$ was considered significant, differences are highlighted. **B** Western blot analysis of Blobby reactivity in adult brain protein extracts from the indicated genotypes, probed with anti-Blobby[ex8b] and anti-Blobby[C-term] antibodies. Anti-Tubulin probing used as a loading control. Prominent Blobby bands are observed in wild type (*w1118*, first lane) but absent in *blobby[Null]* mutant brains. This experiment was performed three independent times with similar results. Source data are provided as a Source Data file. **C** Western blot of

anti-BRP immunoprecipitates from *Drosophila* brain synaptosome preparations, probed for BRP and Blobby[C-term]. Left lane: Input (0,5%), middle lane: gpIgG control, right lane: BRP immunoprecipitate. This experiment was independently repeated twice in quadruplicate (biological replicates). Source data are provided as a Source Data file. **D** Schematic representation of Blobby/CG42795 domain structure, indicating predicted domains: Tre-2, Bub2 and Cdc16 (TBC) domain (green), coiled-coil domains (blue), intrinsically disordered regions (IDR: gray). **E** Representative images of muscle 4 NMJs from third instar control and *blobby[Null]* mutants, labeled with Blobby[C-term] antibody and HRP staining. This experiment was independently repeated two times (each 5 biological replicates) with similar results.

expressing endogenously labeled proteins (BRP labeled with mRuby, Blobby with eGFP; Fig. 2G). In contrast to pr-eexisting AZ scaffolds detected already at the beginning of imaging (at 0 hrs, orange arrowheads), newly forming AZs (white arrowheads) could be identified via their BRP signal at the 24 h timepoint (Fig. 2G, right lower panel). However, these newly formed BRP scaffolds typically lacked Blobby co-localization (see white arrowheads in Fig. 2G left lower panel). Consistent with Blobby's incorporation being delayed relative to BRP, the BRP/Blobby ratio was significantly higher at smaller and thus typically less mature AZs (Fig. 2H).

In summary, Blobby is an additional AZ scaffold protein that incorporates during the late stage of NMJ AZ assembly.

## Reduced active zone numbers at *blobby[Null]* NMJ terminals

We proceeded to investigate the role of Blobby in NMJ development and presynaptic AZ assembly. Thus, we stained larval NMJs for BRP and the postsynaptic glutamate receptor GluRIID (Fig. 3A)[35]. NMJ areas (Fig. 3B) and the total numbers of AZs per NMJ (Fig. 3C; quantified via confocal BRP spot numbers) were both significantly reduced at *blobby[Null]* larval NMJs. At the same time, the AZ density (AZ number normalized to area) was not significantly changed (Fig. 3D).

Upon closer inspection of NMJ BRP distribution, we occasionally observed unusually large BRP aggregates, which we termed "blobs" (Fig. 3A, arrowhead and magnification box). This was the first phenotype we identified, which inspired the naming of the *blobby* mutant.

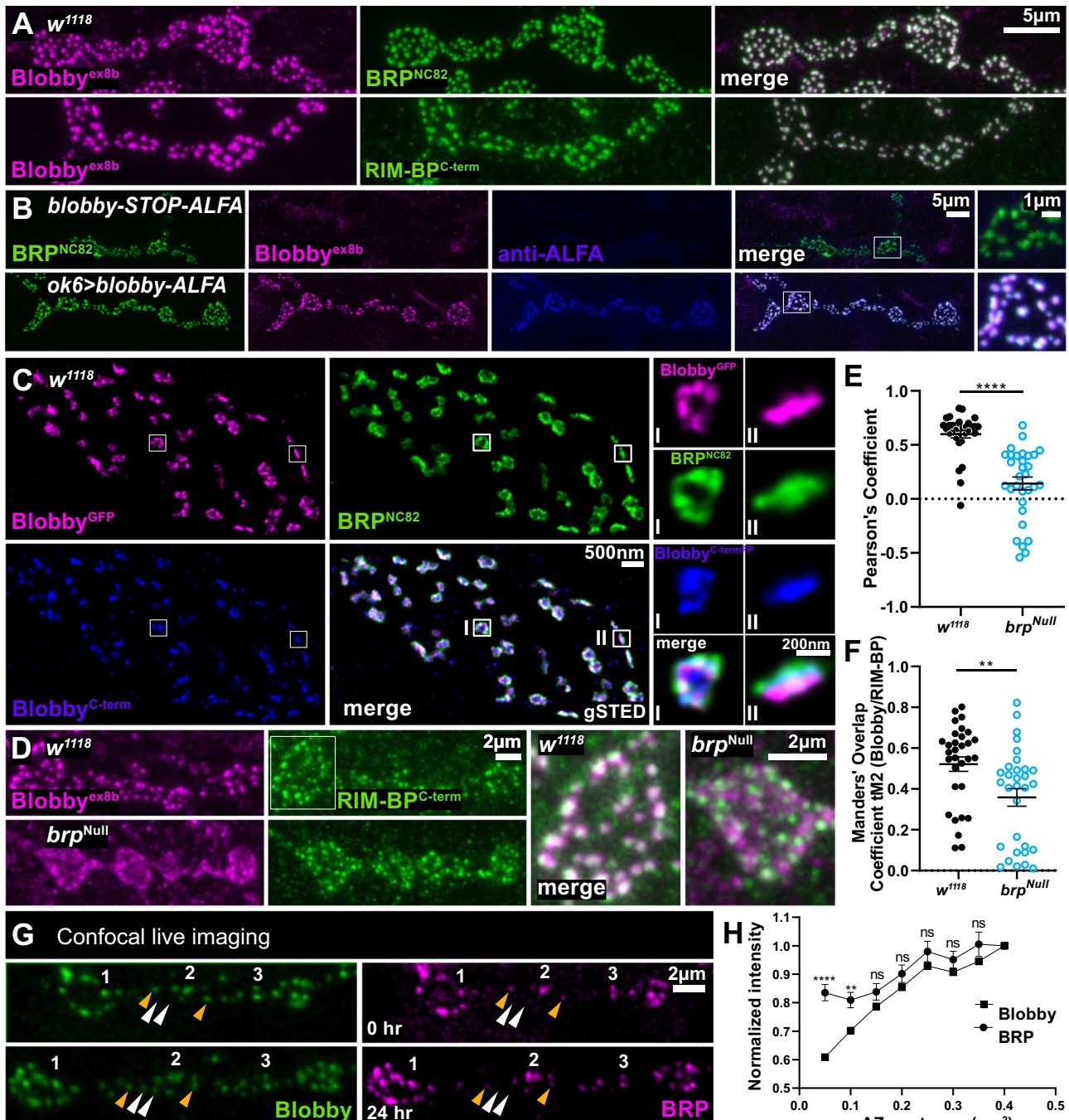

**Fig. 2 | Blobby is integrated into the AZ scaffolds of developing NMJ synapses.**
**A**, **B** Confocal images of muscle 4 NMJs from third instar larvae, labeled with the indicated antibodies. **A** shows wild type (*w1118*) animals, **B** *blobby*-STOP-ALFA and ok6>*blobby-ALFA* (**B**; see main text for genotype details). This experiment was independently repeated twice with 5 biological replicates with similar results. **C** gSTED images of muscle 4 NMJs from larvae expressing endogenously GFP-tagged Blobby (see Fig. S1B), stained for BRP^NC82, Blobby^C-term and GFP. Magnified images show individual AZs in planar (I) and vertical (II) orientations. Measurements: Blobby^GFP - BRP^NC82: 20 ± 2 nm (n = 150 AZs, 4 animals); Blobby^GFP – Blobby^C-Term: 14 ± 2 nm (n = 47 AZs, 4 animals); Blobby^C-Term - BRP^NC82: 17 ± 2 nm (n = 47 AZs, 4 animals). Raw data are provided as a Source Data file.
**D** Representative confocal images and magnifications of muscle 4 NMJs from *w1118* and *brp*^Null mutants, stained with the Blobby^ex8b antibody (magenta) and RIM-BP^C-term (green). **E**, **F** Quantification of overlap between BRP and Blobby^ex8b signals in controls and *brp*^Null. **E** Plot of Pearson´s coefficients: w1118 0.59 ± 0.03, n = 31, N = 4; *brp*^Null 0.14 ± 0.05, n = 31, N = 4, *p* < 0.0001; **F** Plot of Manders coefficients: w1118 0.5 ± 0.04, n = 31, N = 4; *brp*^Null 0.35 ± 0.04, n = 31, N = 4, *p* = 0.0014. Graphs show

mean ± SEM. ****$p < 0.0001$; **$p < 0.01$ (Kolmogorov-Smirnov test was applied). N: number of animals, n: number of boutons. **G** Representative confocal images of muscles 26/27 with Blobby-GFP and BRP-mRuby at 0 hr and 24 hr. The numbers (1,2,3) mark persisting AZs with BRP and Blobby colocalization across the 0 and 24 hr timepoints. Orange arrowheads mark AZ scaffolds where Blobby and BRP colocalized at 0 hr and 24 hr of imaging, white arrowheads mark newly forming AZs with BRP signal (without Blobby) only at the 24 hrs timepoint. **H** Quantification of intensity of BRP and Blobby showing that smaller AZs have higher levels of BRP than Blobby. At 0.05 μm² AZ area, difference between BRP and Blobby 0.2267 ± 0.03978, n = 11, $p < 0.0001$; at 0.1 μm², 0.1077 ± 0.03616, n = 11, $p = 0.0074$ and at 0.15 μm² 0.05137 ± 0.04561, n = 11, $p = 0.2734$; at 0.2 μm², 0,04789 ± 0,04860, n = 11, $p = 0.3362$; 0.25 μm², 0,05053 ± 0,05296, n = 11, $p = 0.9293$; 0.3 μm², 0,04420 ± 0,04320, n = 11, $p = 0.3184$; 0.35 μm², 0,06013 ± 0,06193, n = 11, $p = 0.3432$ Graphs show mean ± SEM. ****$p < 0.0001$; **$p < 0.01$; ns $p > 0.05$. (Unpaired two-sided *t*-test was applied for each AZ spot area). Source data are provided as a Source Data file.

Unlike the typical AZ BRP signals, these blobs were not aligned with the postsynaptic glutamate receptors, as evidenced by the lack of overlay between BRP and GluRIID in confocal image projections (Fig. 3A, arrowhead and box, green blob). Quantification of BRP/GluRIID overlay revealed a significant accumulation of such ectopic material in *blobby^Null* mutants (Fig. 3E; see discussion).

We then similarly immunostained larvae without *(blobby-STOP-ALFA)* and with *(ok6>blobby-ALFA)* motoneuron-specific expression of Blobby (Fig. 3F). Significant reductions of NMJ area (Fig. 3G) and total number of AZs per NMJ (Fig. 3H) were also found in *blobby-STOP-ALFA* relative to *ok6>blobby-ALFA*. The number of AZs per NMJ area was not significantly changed (Fig. 3I). Blobs were also observed in *blobby-STOP-ALFA* but not in *ok6>blobby-ALFA* animals which restore presynaptic Blobby expression (for quantification see Fig. 3J).

In summary, the absence of presynaptic Blobby reduces the number of AZs formed per NMJ terminal. The occasional but robust appearance of ectopic AZ scaffold material suggests that the transfer of BRP into newly forming AZs is inefficient when the BRP-associated protein Blobby is absent (see below and discussion).

## Defective BRP-scaffolds at developing *blobby^Null* active zones

We continued characterizing individual NMJ AZs lacking Blobby. At confocal microscopy resolution, we observed a slight reduction in BRP levels per AZ in *blobby^Null* mutants through two distinct quantification procedures (Fig. S3A, B). We then conducted gSTED analysis. As shown above (Fig. 2C), labeling the BRP C-terminus with the BRP^NC82 monoclonal antibody typically reveals "smooth, ring-like structures" at planar-imaged AZs. However, at *blobby^Null* NMJs, we detected a markedly abnormal pattern in the nanoscale organization and distribution of BRP (compare isogenic controls to *blobby^Null* in Fig. 4A). Triangular and "zig-zag-like" arrays appeared frequently (Fig. 4A, arrows in *blobby^Null*), patterns not observed in controls. Perimeter measurements of BRP clusters at individual AZs, which can distinguish these zig-zag patterns from the typical round appearance of controls, showed significantly elevated values for *blobby^Null* AZs (Fig. 4B).

We then STED imaged scaffold proteins RIM-BP (Fig. 4C) and Unc13A (Fig. 4D), both critical for SV release, whose effective AZ targeting and nano-positioning is organized through binding to the BRP AZ scaffold (see[18,24] for RIM-BP and[8,27] for Unc13A). Given the defective character of the BRP scaffolds, both proteins still formed relatively normal-appearing clusters, with localization patterns mirroring the disturbed BRP nanopattern at *blobby^Null* AZs.

The Ca$^{2+}$ channel α-subunit Cacophony (Cac), evolutionary homologous to the Ca$_v$2.2. family, governs SV release at *Drosophila* NMJ AZs[36], and its proper AZ localization depends on BRP[16,37]. Given the nanoscopic BRP assembly deficits at AZs lacking Blobby, we went on to compare the *blobby* phenotype to the previously analyzed *brp* mutant phenotype. At confocal resolution, Cac clusters appeared per se normal (Fig. S4A), although quantification revealed a slight reduction in intensity (Fig. S4B). STED-resolved Cac Ca$^{2+}$ channel clusters appeared normal in shape and density, and still localized in the AZ centers, surrounded by the ring-like BRP signal in planar-imaged AZs (Fig. 4E). It is worth noting that at BRP-lacking AZs, STED-resolved Cac Ca$^{2+}$ channel clusters are markedly disorganized[37], indicating that the AZ defects of *brp* and *blobby* mutants are qualitatively different.

Since the Cac Ca$^{2+}$ channel clusters mark the center of individual AZs, we took this opportunity to measure AZ-AZ distances (see Material and Methods). The Cac-Cac cluster distances were not increased, again suggesting an essentially unchanged AZ density at *blobby^Null* NMJ terminals (Fig. 4F), consistent with our confocal measurements of AZ density based on BRP signals (Fig. 3D).

Finally, we performed STED analysis on the ectopic large BRP clusters ("blobs") at *blobby* NMJs (Fig. 4G–J). Through co-labeling, we found that these blobs contained RIM-BP (Fig. 4G) and Unc13A (Fig. 4H). In contrast, "early" scaffold protein Syd-1 (Fig. 4I), and "early"

accumulating release factor Unc13B (Fig. 4J) were largely absent from the blobs. This suggests that the blobs may reflect deficits in properly recruiting precursors of "late" scaffold material to nascent AZs (see discussion).

To further investigate their AZ scaffold organization, we subjected *blobby^Null* NMJ terminals to transmission electron microscopic analysis (EM). Using standard protocols, when glutaraldehyde fixation is employed, wild type *Drosophila* presynaptic AZ membranes exhibit electron-dense structures known as T-bars, which represent the structural imprint of the BRP-organized AZ scaffold[38,39] (Fig. 5A, arrow labels T-bar "roof", arrowhead indicates the T-bar "pedestal"). Given that BRP is a major, essential constituent of the T-bar[37], irregularities in BRP nanoarchitecture are expected to affect T-bar organization. Visual inspection and quantification of *blobby* mutants (Fig. 5B) revealed a near-complete absence of properly shaped T-bars, with irregular electron-dense structures observed in their place (arrowhead, Fig. 5A), consistent with the disturbed BRP nanopattern observed under STED (Fig. 4A).

Overall, Blobby plays a crucial role in establishing the precise nanoscale organization of the BRP-scaffold at developing NMJ AZs, as evidenced by both STED and EM.

## Evoked synaptic vesicle release is inefficient at NMJ active zones lacking Blobby

The BRP-scaffold serves roles in the AZ membrane targeting and docking of SVs[24,40]. To visualize SVs, we used high-pressure freeze electron microscopy (HPF-EM) to preserve the physiological state of the biological samples[41–43]. In *Drosophila*, SVs are smaller than 40 nm in diameter, while larger, clear vesicles dominantly reflect endosomal sorting vesicles[44]. Upon visual inspection, the distribution of SVs smaller than 40 nm and larger vesicles exceeding 40 nm appeared similar between controls and *blobby* mutant AZs and boutons (Fig. 5C).

SVs that were close to or physically attached to the AZ plasma membrane ("docked", Fig. 5C, arrowheads) are considered biochemically primed and ready for release. When counting SVs (diameter <40 nm) within 5 or 10 nm of the inner leaflet of the AZ membrane, we observed no significant differences between *blobby^Null* and control AZs (Fig. 5D, E). Additionally, we analyzed the distributions of vesicles in both size classes (<40 nm, Fig. 5F; >40 nm, Fig. 5G, Fig. S5) as a function of their distance from the AZ plasma membrane. To quantify vesicle distributions, we focused on vertically imaged AZ regions, identified by the planar membrane contact between the presynaptic motoneuron and the postsynaptic muscle plasma membrane (Fig. 5C boxes). Vesicles were counted within defined rectangular areas, measuring 400 nm horizontally along the AZ membrane and 450 nm vertically. Overall, vesicle distribution profiles appeared similar between controls and *blobby^Null*, though there was a tendency for an increased number of larger clear vesicles in *blobby^Null* at positions further from the AZ membrane (Fig. 5G).

We next evaluated Blobby's impact on SV release at larval NMJ synapses and performed two-electrode voltage clamp (TEVC) recordings to measure both action potential-evoked excitatory junctional currents (eEJCs) and spontaneous miniature SV release (mEJCs). We again compared two genetic scenarios involving the loss of presynaptic Blobby: *blobby^Null* versus its isogenic control (Fig. 6A–H), as well as *ok6,blobby-STOP-ALFA* (mutant) versus the corresponding control, *ok6>blobby-ALFA* (Fig. 6I–P).

Action potential-evoked responses (eEJCs) were significantly reduced at NMJs lacking Blobby (Fig. 6A, B, I, J). Additionally, mEJC amplitudes were significantly reduced relative to controls in *blobby^Null* animals (Fig. 6G). Quantal content, which indicates the number of SVs released per action potential, was calculated by dividing eEJC values by the respective miniature excitatory junctional amplitudes. Both Blobby-deficient genotypes displayed a strong and significant reduction in quantal content (Fig. 6C–K).

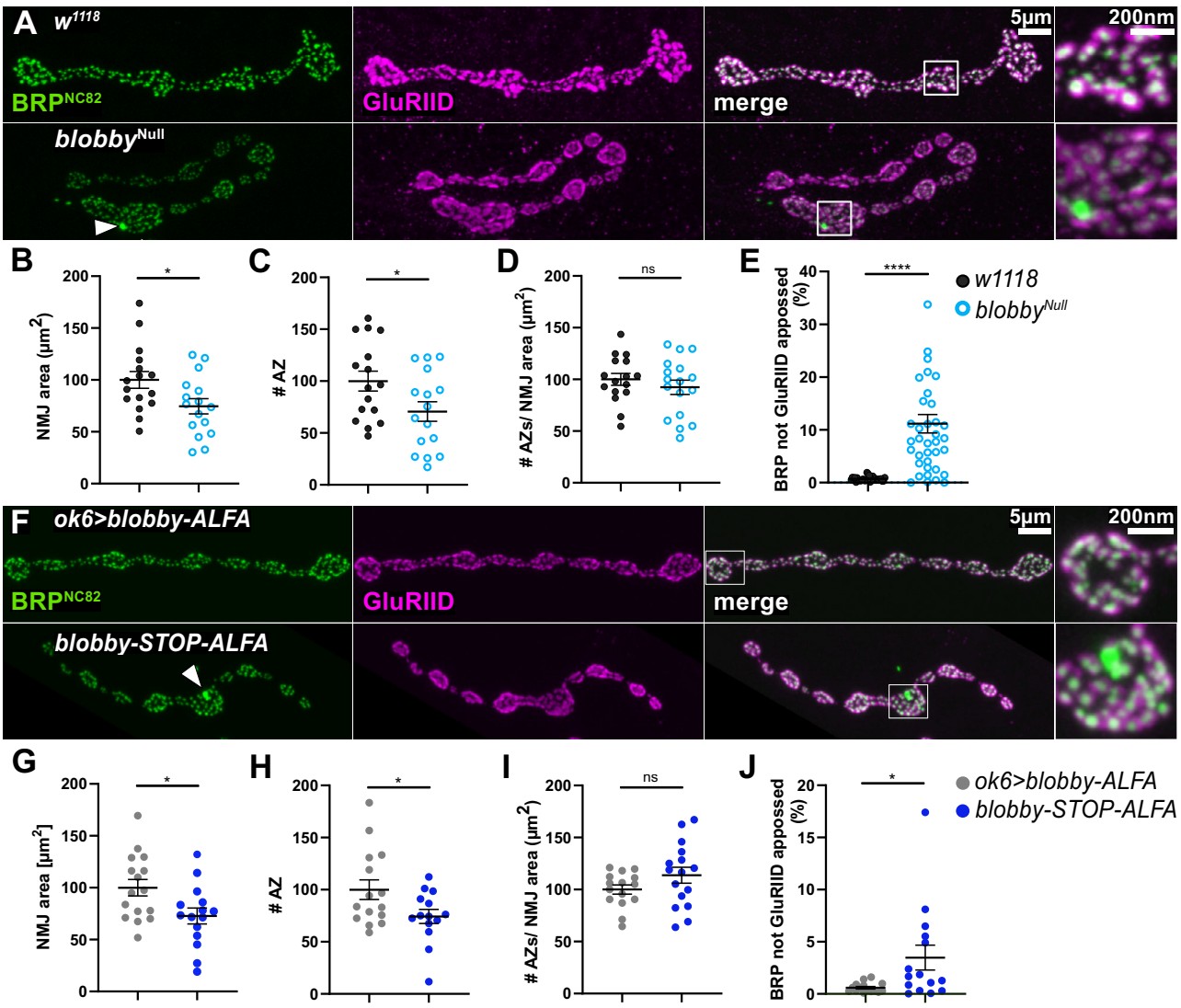

**Fig. 3 | Analysis of *blobby* mutant NMJs via confocal microscopy. A–J:** Confocal imaging analysis comparing larval NMJs of *blobby*^Null^ animals to *w1118* controls (**A–E**) and *blobby-STOP-ALFA* to ok6>*blobby-ALFA* (**F–J**) larvae. **A, F** Representative confocal images of muscle 4 NMJs of third instar larvae stained with BRP^NC82^ (green) and GluRIID (magenta) of indicated genotypes (**B–E**) and (**G–J**): Quantification of NMJ morphology and AZ numbers of indicated genotypes. **B, G** Normalized NMJ area (in μm²): *w1118* 100 ± 8.1, n = 16, N = 4; *blobby*^Null^ 74.58 ± 7.26, n = 16, N = 4, *p* = 0.0263; ok6>*blobby-ALFA* 100 ± 7.9, n = 16, N = 4; *blobby-STOP-ALFA* 72.78 ± 7.66, n = 15, *N* = 4, *p* = 0.0197; unpaired student two-sided *t*-test; (**C, H**) AZs identified as discrete BRP positive clusters per NMJ normalized to controls: *w1118* 100 ± 9,6 n = 16, N = 4; *blobby*^Null^: 70.7 ± 9.4, n = 16, N = 4, *p* = 0.0368; ok6>*blobby-ALFA*: 100 ± 9.49, n = 15, N = 4; *blobby-STOP-ALFA* 74.29 ± 6.78, n = 14, N = 4,

*p* = 0.0385 unpaired student two-sided *t*-test. **D, I** Number of AZs normalized to NMJ area from projected images, values normalized to respective controls: *w1118* 100 ± 5.71, n = 16, N = 4; *blobby*^Null^ 92.35 ± 6.89, n = 17, N = 4, *p* = 0.4023; ok6>*blobby-ALFA* 100 ± 4.35, n = 15, N = 4; *blobby-STOP-ALFA* 113.7 ± 7.7, n = 16, N = 4, *p* = 0.1384 unpaired student two-sided *t*-test. **E, J** Quantification of ectopically distributed scaffold material as BRP signal not apposed to GluRIID signals in projected confocal images; *blobby*^Null^ in comparison to control: *w1118* 0.7% ± 0.07%, n = 32, N = 5; *blobby*^Null^ 11.2% ± 1.7%, n = 38, N = 5, *p* < 0.0001; ok6>*blobby-ALFA* 0.59% ± 0.1%, n = 15, N = 4; *blobby-STOP-ALFA* 3.5% ± 1.2%, n = 15, N = 4, *p* = 0.0281, Kolmogorov-Smirnov test. Graphs show mean ± SEM. *****p* < 0.0001; **p* < 0.05. N: number of animals, n: number of NMJs analyzed. Source data are provided as a Source Data file.

Facilitation or depression, in response to rapid presynaptic stimulation, is commonly used to estimate SV release probability, with increased facilitation indicating a decreased release probability. Paired-pulse ratios were significantly elevated across two interpulse intervals in both the *blobby*^Null^ compared to the isogenic control and *ok6,blobby-STOP-ALFA* versus *ok6>blobby-ALFA* conditions (Fig. 6D, E, L, M). This suggests that a decrease in SV release probability contributes to the release deficits observed in Blobby-lacking AZs, alongside a loss of evoked release which should result from their reduced physical numbers of AZs per NMJ.

Concerning the timing of evoked release, *blobby*^Null^ animals showed increased eEJC rise times (Fig. S6 A) and decay times (Fig. S6 B). mEJC rise (Fig. S6 C) and decay times (Fig. S6 D) were increased as

well. The *ok6,blobby-STOP-ALFA* animals (Fig. S6 F–J) showed elevated rise times of spontaneous release (Fig. S6 H) relative to the corresponding controls. This difference might be explained by differences in the genetic background of both mutants, *blobby*^Null^ and *ok6,blobby-STOP-ALFA* larvae.

Since Blobby was identified through its biochemical association and close colocalization with BRP, we investigated the extent to which the phenotypes of *blobby*^Null^ and *brp* mutants might resemble each other. In *brp* mutants, the residual SV release is characterized by supersensitivity to the slow Ca²⁺ buffer EGTA, which suggests an increased coupling distance between Cac Ca²⁺ channels and SVs[37]. To explore whether this would be the case for *blobby* as well, we TEVC measured both evoked (Fig.S7 A–G) and spontaneous release (Fig. S7 H–L) in

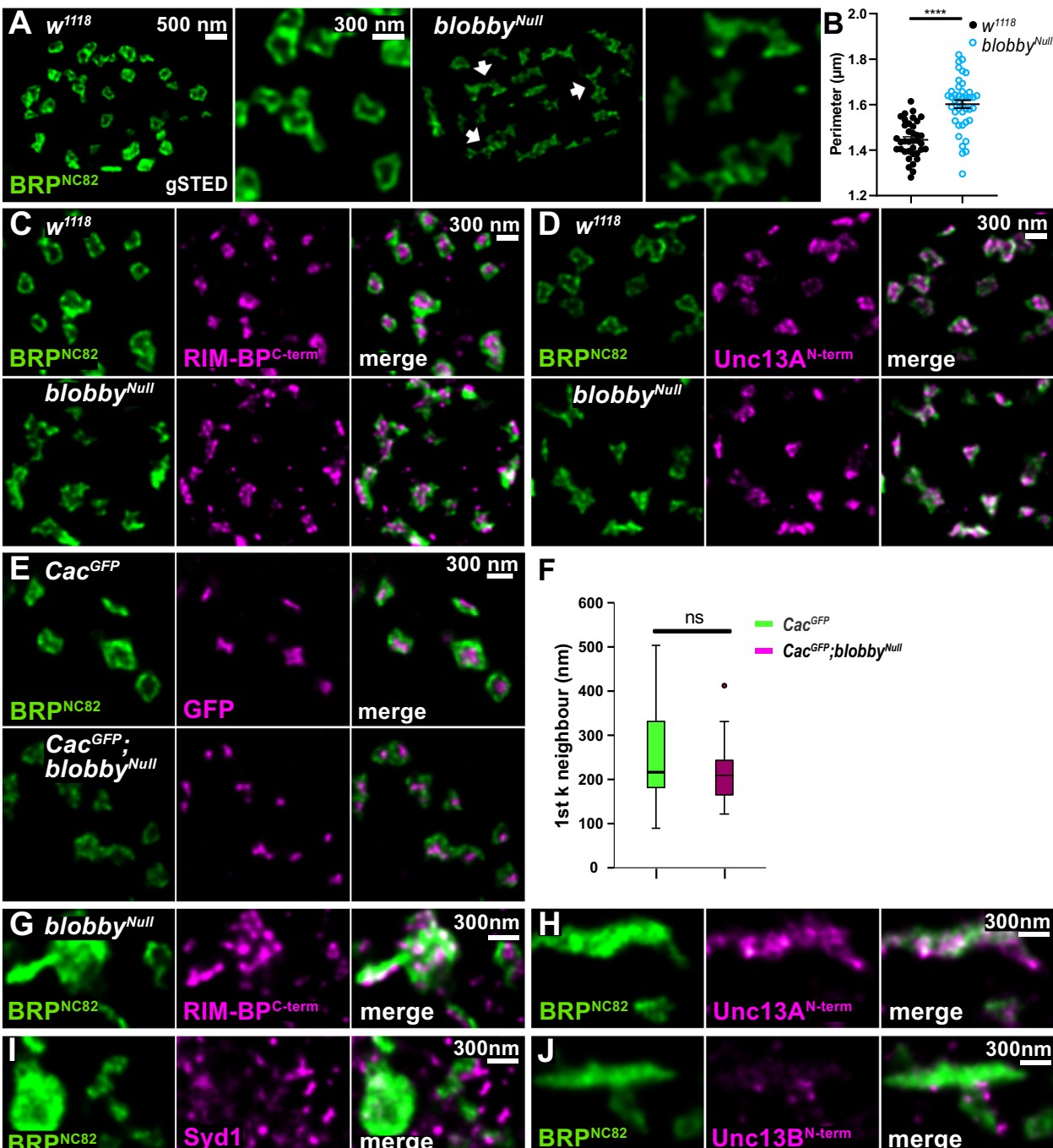

**Fig. 4 | gSTED characterization AZ scaffolds and ectopic scaffold accumulations ("blobs") in _blobby^Null_. A** gSTED images of muscle 4 neuromuscular junctions of _w1118_ and _blobby^Null_ AZs stained for BRP. Arrows indicate disrupted "zig-zag-like" BRP scaffolds of _blobby^Null_ mutants. Scale Bar of overview bouton: 500 nm. Scale Bar of magnified images of individual AZs: 300 nm. **B** Quantification of AZ nanoarchitecture: perimeters of BRP gSTED images from _w1118_ and _blobby^Null_ were analyzed (_w1118_ 1.40 ± 0.01, n = 37; _blobby^Null_ 1.60 ± 0.02, n = 42, p < 0.0001. Graph shows mean ± SEM. Data distribution is normal according to the D'Agostino and Pearson omnibus normality test. Unpaired two-sided _t_-test was applied****p < 0.0001. N: number single boutons analyzed in gSTED. **C, D** gSTED images of muscle 4 NMJs of _w1118_ and _blobby^Null_ AZs co-stained for BRP (green), (**C**) RIM-BP (magenta) and (**D**) Unc13A (magenta). Scale Bar: 300 nm. **E** gSTED images of muscle 4 NMJs of on-locus GFP-labeled Cacophony Ca$^{2+}$ channel (Cac^GFP) in control and _blobby^Null_ background stained for BRP (green) and GFP (magenta). Scale Bar: 300 nm. **F** Analysis of nearest neighbor inter- sfCac^GFP distances. Presynaptic CacGFP clusters within 1 μm radius from a given Cac^GFP cluster were determined. SfCac^GFP 252.52 ± 17.09, n = 43 sfCac^GFP; _blobby^Null_ 214.27 ± 11.54, n = 30. Graphs shows median, lower and upper quartiles, whiskers represent min/max score s. Circles are outliers. Mean ± SEM values. Mann Whitney _U_ Test was performed, ns = not significant. **G–J** gSTED images of individual "blobs" at _blobby^Null_ NMJs stained for BRP (green) together with (**G**) RIM-BP (magenta), (**H**) Unc13A (magenta), (**I**) Syd-1 (magenta), (**J**) Unc13B (magenta). Scale Bar: 300 nm. The experiments (**C–J**) were performed three and (**I**) two independent times with similar results. Source data are provided as a Source Data file.

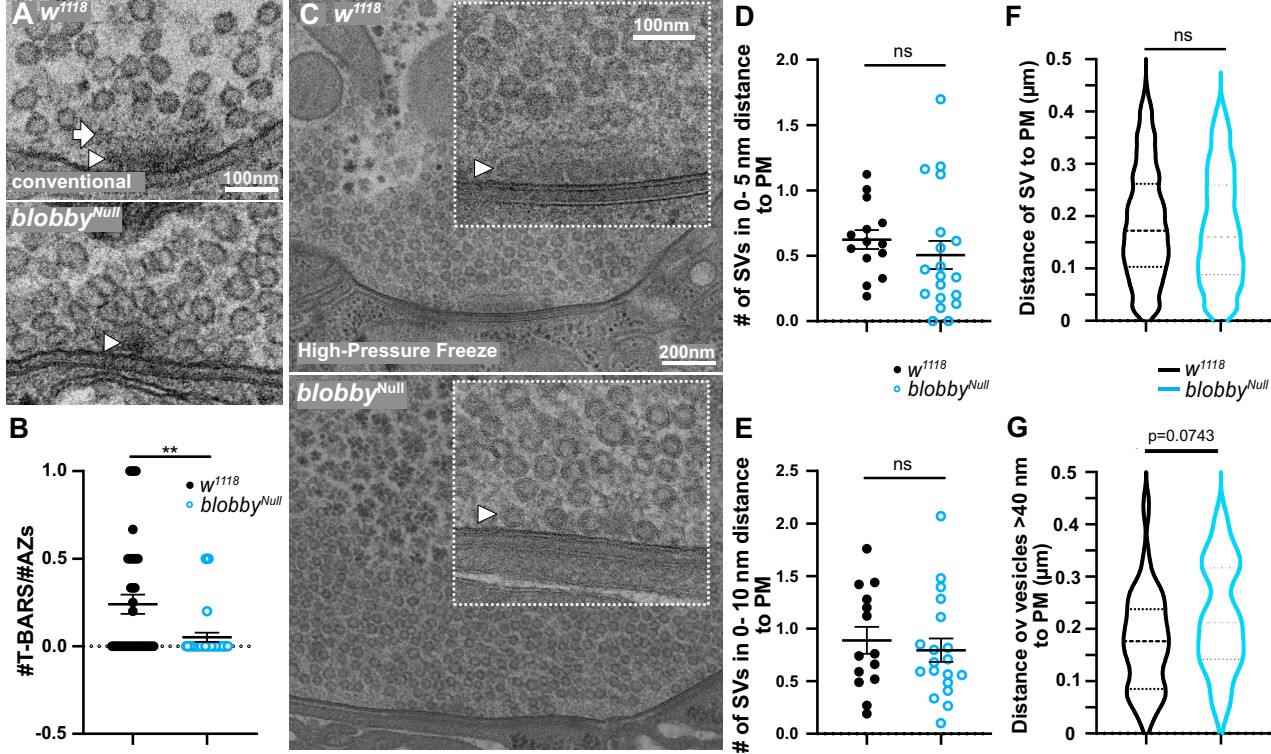

**Fig. 5 | Electron microscopic analysis of *blobby* mutant NMJs.** Representative ultrastructural images of control (*w1118*) and *blobby*^Null mutant boutons from (**A-B**) conventional standard transmission using glutaraldehyde fixation or (**C–G**) high-pressure freeze (HPF) prepared NMJs. **A** Arrow labels T-Bar roof, arrowhead T-Bar pedestal. Scale Bar: 100 nm. **B** Quantification of morphologically identified T-Bars per AZ section (*w1118* 0.24 ± 0.05, n = 38, N = 80; *blobby*^Null 0.05 ± 0.02, n = 33; N = 76, *p* = 0.0039). Graph shows mean ± SEM. n: number of boutons; N: number of AZs. 3 animals per genotype were analyzed. A two-sided Mann-Whitney test was applied. **C** HPF images of representative half boutons and magnified views of individual active zones (AZs). Arrow head points towards docked SVs. **D-E** Quantification of synaptic vesicles (SVs) profiles (< 40 nm) within 0–5 (*w1118*

0.62 ± 0.07, n = 14; *blobby*^Null 0.51 ± 0.11, n = 19, *p* = 0.1207) (**D**) or 0 -10 nm (**E**) nm from the AZ plasma membrane per 400 nm AZ (*w1118* 0.88 ± 0.12, n = 14; *blobby*^Null 0.79 ± 0.10, n = 19, *p* = 0.8381). Graph shows the mean ± SEM. n: number of boutons from 3 animals per analyzed genotype. Kolmogorov-Smirnov test was applied. **F-G** Quantification of AZ plasma distance distribution of (**F**) SVs *w1118* 0.18 ± 0.003, *n* = 702; *blobby*^Null 0.18 ± 0.004, *n* = 497, *p* = 0.1042, Mann-Whitney test was applied. **G** larger, clear vesicles (>40 nm) (**G**): *w1118* 0.18 ± 0.02, n = 30; *blobby*^Null 0.22 ± 0.02, n = 35, *p* = 0.0743, unpaired two-sided *t*-test was applied. Graph shows the mean ± SEM. n: number of boutons from 3 animals per analyzed genotype. Source data are provided as a Source Data file.

*blobby*^Null mutants using the same protocol as previously used for *brp* mutants, involving a 30-min EGTA-AM ester preincubation[37]. However, in *blobby*^Null mutants, neither the efficacy (Fig. S7 A, B), nor the temporal structure (Fig. S7 C, D) or the paired pulse behavior of evoked release (Fig. S7 F, G) showed increased sensitivity to EGTA relative to controls. This result largely rules out differences in the coupling between Ca²⁺ channels and SV Ca²⁺ sensors as the primary cause of the release deficits at Blobby-lacking NMJ synapses, consistent with the apparently largely intact clustering and cluster spacing of Unc13A (Fig. 4D) and voltage-gated Ca²⁺ channels (Fig. 4E).

In summary, we found no evidence of deficits in the nano-spacing between Ca²⁺ channels and SV release sites or in the targeting of SVs to the AZ plasma membrane as contributors to the functional deficits in *blobby* mutants. Instead, our data suggest that a reduction in the total number of AZs per NMJ terminal (Fig. 3C), along with a decreased SV release probability, indicated by the robustly increased paired-pulse responses (Fig. 6D, E; 6L, M; Fig. S7 F, G), accounts for the observed release deficits.

### Reduced release probability and release site number drive the *blobby* release deficit

To further support this interpretation physiologically, we conducted a mean-variance analysis, which takes advantage of the probabilistic nature of vesicle fusion (see Methods). In this approach, evoked excitatory junctional currents (eEJCs) were measured across a range of

extracellular Ca²⁺ concentrations (Fig. 7A), and the variance of eEJC amplitudes was plotted against the eEJC amplitude (Fig. 7B). Low variance is typically observed at both low and high (saturating) Ca²⁺ concentrations. Fitting a parabola to the data (Fig. 7B) allows to extract key parameters related to synaptic release. The quantal size of evoked release was not significantly altered (Fig. 7C). However, we observed a reduction in SV release probability at physiological Ca²⁺ concentrations of 1.5 mM (same concentration as used in Fig. 6), based on mean-variance analysis (Fig. 7D). This deficit could be overcome by higher Ca²⁺ concentrations. A Hill curve plot (Fig. 7E) indicated that the Ca²⁺ cooperativity of SV release remained essentially normal in *blobby* mutants. Notably, the estimated number of functional release sites per NMJ (N) was significantly reduced (Fig. 7F), which aligns with the reduction in the physical number of AZ release sites observed (Fig. 3C).

Taken together, mean-variance analysis confirms that both, a reduced number of SV release sites but also deficits in SV release probability underlie the release deficits of *blobby* mutants at physiological Ca²⁺ concentrations.

### Blobby is essential for the consolidation of olfactory memories in the mushroom body

We finally explored whether Blobby function was also relevant in the adult *Drosophila* brain and whether it might potentially be important for behavior. The *blobby*^Null animals survived to adulthood, albeit only in significantly reduced numbers (Fig. 8A). The locomotion of the

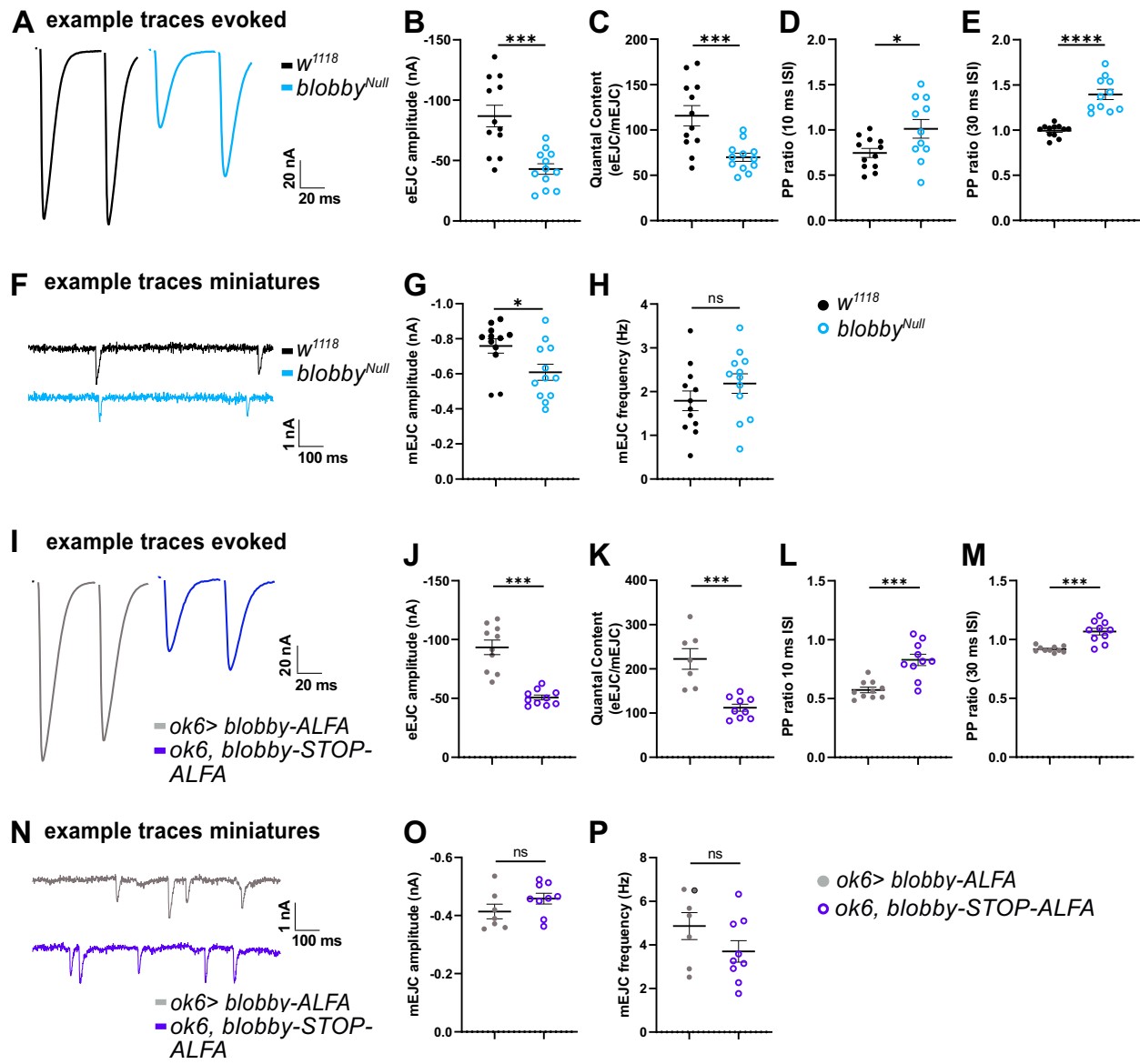

**Fig. 6 | Two-electrode voltage clamp analysis of *blobby* mutant NMJs. A–P** Two-electrode voltage clamp electrophysiological recordings comparing third instar larvae NMJs of *blobby*^Null^ animals to controls (**A–H**) and *ok6, blobby-STOP-ALFA* to ok6>*blobby-ALFA* (**I-P**). **A, I** Example traces of evoked responses (eEJC) at interpulse interval of 30 ms. **B, J** eEJC amplitudes (*w1118* -86.91 ± 8.96 nA, n = 12; *blobby*^Null^ -43.03 ± 4.40 nA, n = 12, *p* = 0.0002; *ok6*>*blobby-ALFA* -93.42 ± 6.13 nA, n = 10; *ok6,blobby-STOP-ALFA* -45.73 ± 3.40 nA, n = 10, *p* < 0.0001). **C, K** Quantal contents (*w1118* 115.70 ± 11.09, n = 12; *blobby*^Null^ 69.88 ± 4.49, n = 12, *p* = 0.0009; *ok6*>*blobby-ALFA* 222.20 ± 23.23, n = 7; *ok6, blobby-STOP-ALFA* 87.74 ± 6.96, n = 10, *p* = 0.0002). **D, L** Paired-pulse ratio at 10 ms interpulse interval. (*w1118* 0.74 ± 0.05, n = 12; *blobby*^Null^ 1.01 ± 0.10, n = 11, *p* = 0.0269; *ok6*>*blobby-ALFA* 0.57 ± 0.02, n = 10; *ok6,blobby-STOP-ALFA* 1.03 ± 0.05, n = 10, *p* = 0.0002). **E, M** Paired-pulse ratio at

30 ms interpulse interval. (*w1118* 0.99 ± 0.02 n = 12; *blobby*^Null^ 1.39 ± 0.06, n = 11, *p* < 0.0001; *ok6*>*blobby-ALFA* 0.92 ± 0.01, n = 10; *ok6,blobby-STOP-ALFA* 1.30 ± 0.02, n = 10, p < 0.0001). **F, N** Example traces of miniature responses. **G, O** mEJC (miniature) amplitudes (*w1118* -0.75 ± 0.04 nA, n = 12; *blobby*^Null^ -0.60 ± 0.05 nA, n = 12, *p* = 0.0228; *ok6*>*blobby-ALFA* -0.41 ± 0.03 nA, n = 7; *ok6,blobby-STOP-ALFA* -0.55 ± 0.05 nA, n = 10, *p* = 0.1658). **H, P** mEJC frequencies (*w1118* 1.79 ± 0.22, n = 12; *blobby*^Null^ 2.18 ± 0.22, n = 12, *p* = 0,2261; *ok6*>*blobby-ALFA* 4.83 ± 0.62, n = 7; *ok6,blobby-STOP-ALFA* 1.66 ± 0.31, n = 10, *p* = 0.1663). Graphs (**B–E,G,H, J–M,O,P**) show mean ± SEM. An unpaired two-sided *t*-test was applied, *\*p* < 0.05; \*\*\**p* < 0.001; \*\*\*\**p* < 0.0001 ns = not significant. *n* represents a single cell. Four to six animals are analyzed with one or two cells/animal. Source data are provided as a Source Data file.

surviving *blobby*^Null^ adult animals was significantly reduced (Fig. 8B). In adult brains, Blobby antibodies labeled the synaptic neuropil but not the cell body regions throughout the entire *Drosophila* brain (Fig. 8C), showing extensive co-localization with BRP. In *blobby*^Null^ brains, Blobby antibody staining was absent (Fig. 8C, lower row), proving antibody specificity in this tissue and that Blobby is present in the post-developmental adult *Drosophila* brain. We observed that Blobby levels were particularly high at the axonal lobes of α/β neurons (Fig. 8C), a subclass of the mushroom body intrinsic neurons (Kenyon cells) crucial for forming consolidated forms of olfactory memory[45].

Recent studies have shown that AZ plasticity involving the BRP-based scaffold plays a specific role in the post-conditioning consolidation ("mid-term memory, MTMs") of aversive olfactory memories within Kenyon cells measured hours after conditioning, while it is dispensable for short-term memory (STM, "learning") measured after minutes[46]. This consolidation-specific role of BRP was demonstrated via post-developmental knock-down (KD) of BRP specifically within Kenyon cells. Thus, to avoid interfering with circuit and synapse development, post-developmental KD of Blobby (using a timed KD induced via a temperature shift which inactivates the *Gal80*^ts^

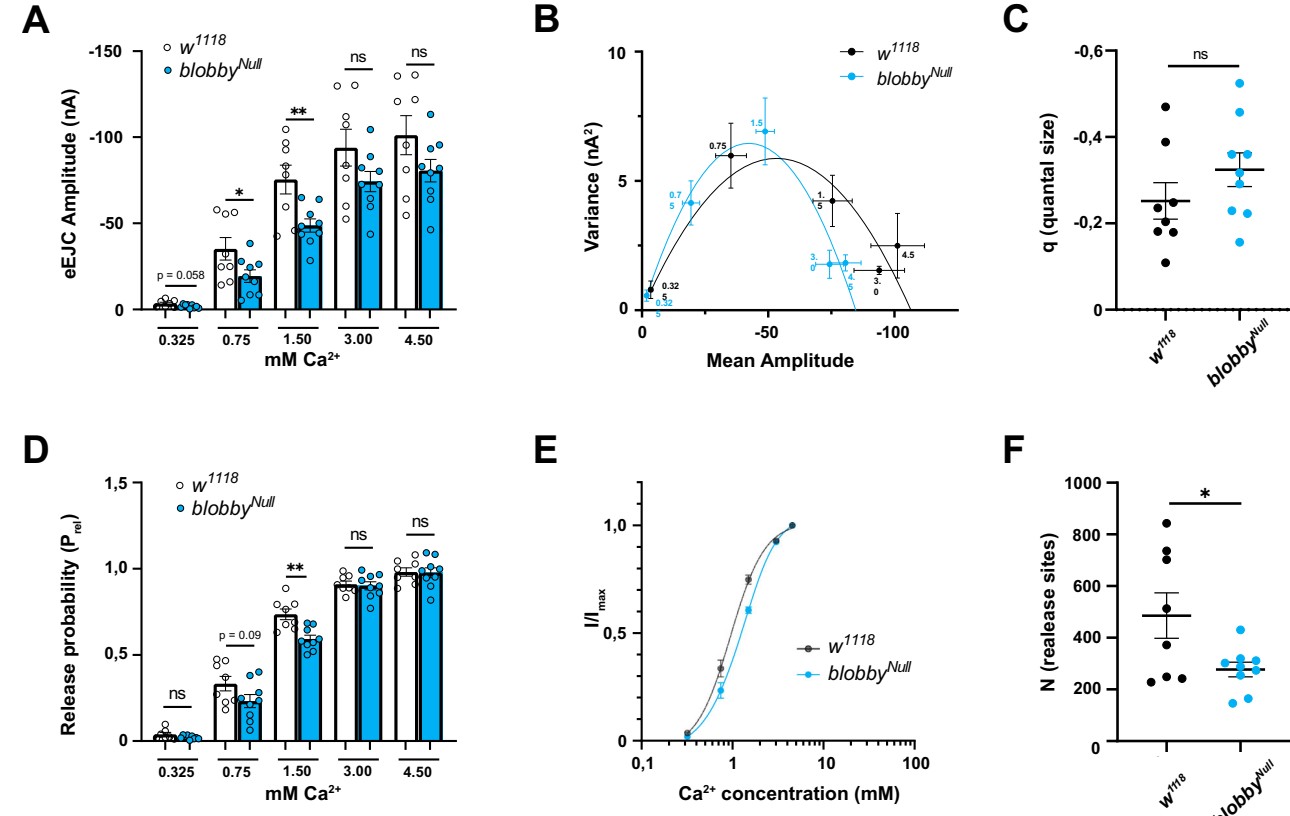

**Fig. 7 | Mean-variance analysis of SV release in *blobby* mutants. A–F** Two-electrode voltage clamp electrophysiological recordings comparing third instar larvae NMJs of *blobby^Null* animals to controls. **A** Evoked excitatory junctional currents (eEJCs) measured across a range of extracellular Ca²⁺ concentrations, highlighting the relationship between calcium and synaptic response (0.325 mM: *w1118* -5.36 ± 1.78 nA, n = 8, *blobby^Null* -2.20 ± 0.43 nA, n = 6, *p* = 0.0578; 0.75 mM: *w1118* -33.56 ± 4.47 nA, n = 8, *blobby^Null* -17.76 ± 4.79 nA, n = 6 *p* = 0.0455; 1.5 mM: *w1118* -70.40 ± 7.52 nA, n = 8, *blobby^Null* -49.62 ± 3.26 nA, n = 6 *p* = 0.0086; 3.0 mM: *w1118* -96.19 ± 8.07 nA, n = 8, *blobby^Null* -71.53 ± 3.25 nA, n = 6, *p* = 0.1169; 4.5 mM: *w1118* -105.00 ± 8.47 nA, n = 6, *blobby^Null* -79.81 ± 4.67 nA, n = 6, *p* = 0.1258). **B** Variance of eEJC amplitudes plotted against the eEJC amplitude. A parabolic fit to the data allows for extraction of key parameters related to synaptic release, showing typical low variance at both low and high (saturating) Ca²⁺concentrations. n = 1 animal. **C** Quantal size of evoked release is shown to remain unaltered in blobby mutants (*w1118* -0.25 ± 0.04 pC, n = 8; *blobby^Null* -0.32 ± 0.04 pC, n = 9, *p* = 0.2267). **D** Release

probability of synaptic vesicles (SVs) at physiological Ca²⁺concentration (1.5 mM) is reduced in *blobby* mutants, as revealed by mean-variance analysis. This deficit is overcome at higher Ca²⁺calcium concentrations (0.325 mM: *w1118* 0.04 ± 0.01, n = 7, *blobby^Null* 0.02 ± 0.003 nA, n = 9 *p* = 0.1472; 0.75 mM: *w1118* 0.33 ± 0.04, n = 8, *blobby^Null* 0.23 ± 0.04, n = 9 *p* = 0.0920; 1.5 mM: *w1118* 0.74 ± 0.03, n = 8, *blobby^Null* 0.59 ± 0.02, n = 9 *p* = 0.0016; 3.0 mM: *w1118* 0.91 ± 0.02, n = 8, *blobby^Null* 0.90 ± 0.23, n = 9 *p* = 0.7497; 4.5 mM: *w1118* 0.98 ± 0.02, n = 8, *blobby^Null* 0.97 ± 0.03 nA, n = 9 *p* = 0.9027). **E** Hill curve plot demonstrating that the Ca²⁺ cooperativity of SV release remains normal in blobby mutants. **F** The estimated number of functional release sites per NMJ (**N**) is significantly reduced in *blobby* mutants. (*w1118* 316.90 ± 27.56, n = 8; *blobby^Null* 146.20 ± 28.35, n = 6, *p* = 0.0314). Graphs show mean ± SEM. An unpaired two-sided *t*-test was applied, *\*p* < 0.05; *\*\*p* < 0.01; *\*\*\*\*p* < 0.0001 ns = not significant. n represents a single cell. Four to six animals were analyzed with one or two cells/animal. Source data are provided as a Source Data file.

---

suppressor) was established. Our temperature shift protocol when using a pan-neuronal driver line reduced Blobby protein levels by about 50% (not shown). This post-developmental Blobby KD targeted to all Kenyon cells did not affect initial learning, i.e. the acquisition of STMs (Fig. 8D). In contrast, however, it nearly eliminated the consolidation of MTM at 1 h (Fig. 8E, low temperature control in Fig. 8F) and at 3 h (Fig. 8G, low temperature control in Fig. 8H) post-conditioning. Thus, even a moderate post-developmental KD of *blobby* within the mushroom body Kenyon cells is sufficient to effectively disrupt behaviorally relevant AZ functionalities.

In summary, the AZ-scaffold protein Blobby appears to be critical for proper AZ assembly during development, but also plays a significant role at mature AZs in supporting behavior.

## Discussion

We here report the identification and phenotypic characterization of Blobby, an additional AZ scaffold localized protein at pre-synapses of *Drosophila* synapses. STED microscopy analysis using specific antibodies revealed that Blobby is confined to the core AZ scaffolds at

larval NMJs. Intravital imaging found Blobby to integrate into nascent NMJ AZs together with ELKS-family scaffold protein BRP, and Blobby was detected in BRP immuno-precipitates. Additionally, incorporation of Blobby into AZ scaffolds depended on BRP. Importantly, the loss of Blobby resulted in fewer AZs forming per terminal, though the AZ density at the smaller NMJ terminals did not decrease. Likely, the loss of AZs forming contributes directly to the reduction of evoked SV release in this mutant.

The absence of Blobby led to the formation of ectopic BRP aggregates ("blobs"), which contained late AZ scaffold proteins like BRP and RIM-BP but lacked early scaffold proteins such as Liprin-α and Syd-1. Intravital imaging of *Drosophila* NMJ AZs revealed that BRP accumulation is a gradual process that unfolds over several hours[32–34]. Our findings show that Blobby incorporates late into forming AZ scaffolds (Fig. 2G), suggesting it plays a crucial role in integrating BRP/RIM-BP precursor material. The formation of ectopic blobs may result from improper integration of scaffold material at nascent AZs. We speculate that Blobby may facilitate the conversion of BRP/RIM-BP precursor material into a "consumable state," potentially by exposing

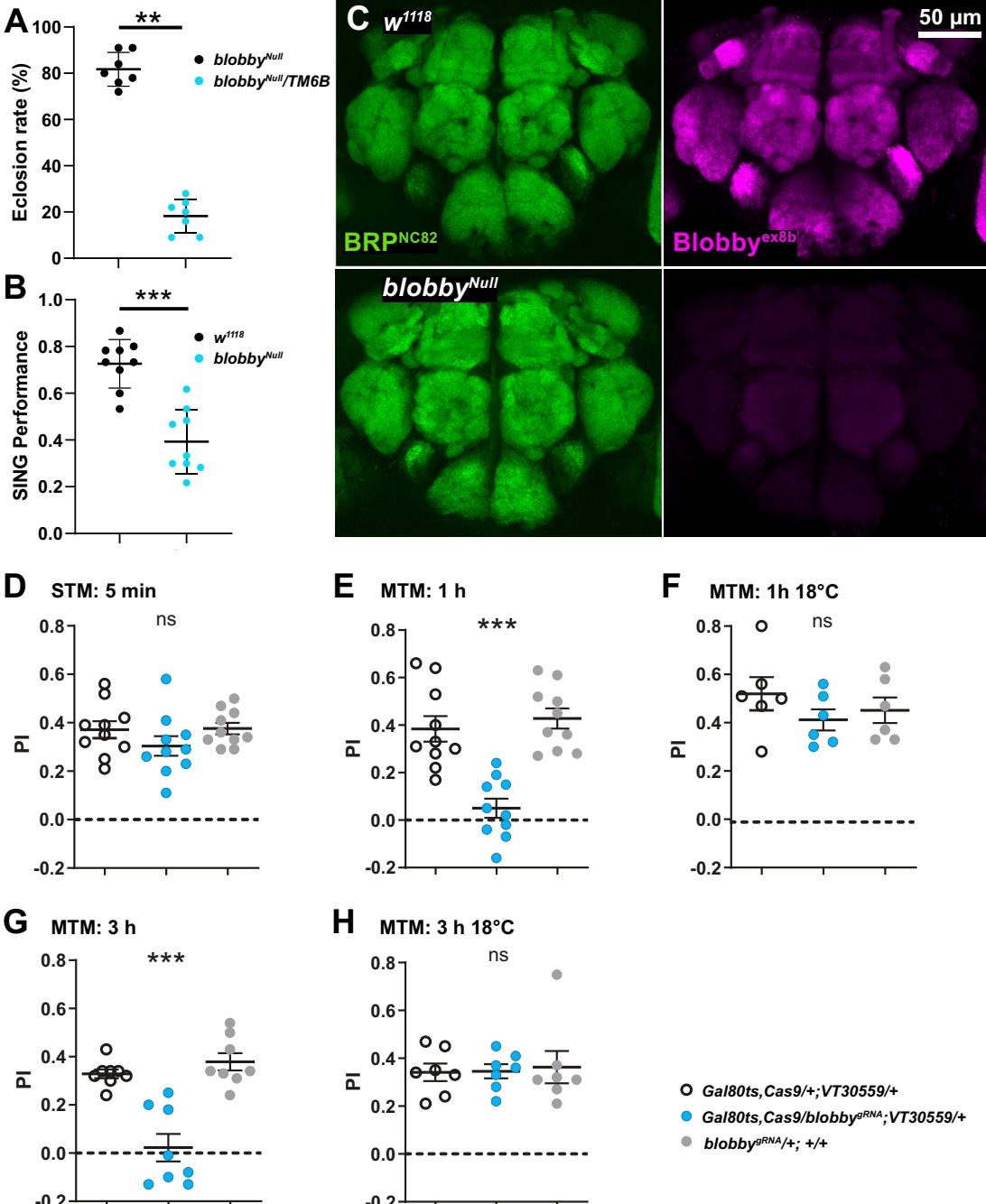

**Fig. 8 | Blobby in the adult *Drosophila* brain and behavior. A** Eclosion rates of *blobby^Null* homozygous adult flies compared to heterozygous controls: *blobby^Null* adults hatched at significantly lower rates than the expected 33%. (*blobby^Null* /TM6B 81.71 ± 2.766, n = 7; *blobby^Null* 18.29 ± 2.766, n = 7. Chi-square test: Chi-square = 21.51, df = 6, *p* = 0.0015). Graphs show mean ± SD. **B** Startle induced negative geotaxis (SING) test to estimate locomotion speed. *blobby^Null* animals are significantly reduced in their locomotive response. (*w1118* 0.7259 ± 0.03448, n = 9, *blobby^Null* 0.3926 ± 0.04523, n = 9, *p* < 0.0001, unpaired two-sided *t*-test was applied). Graphs show mean ± SD. **C** Immunostaining of adult *Drosophila* brains using Blobby antibodies reveals strong labeling in the synaptic neuropil, whereas no labeling is observed in cell body regions. High Blobby expression is detected in the axonal lobes of α/β neurons, a subclass of mushroom body neurons involved in olfactory memory consolidation. Blobby staining is absent in *blobby^Null* brains, confirming antibody specificity and Blobby's presence in the post-developmental adult brain.

**D–H** Post-developmental knockdown (KD) of *blobby* in Kenyon cells does not affect the acquisition of (**D**) short-term memories (STM) *p* = 0.0.0574, n = 10, but significantly and severely impairs the consolidation of mid-term memories (MTM) at both (**E**) 1 h F(3,30) = 20.23, *p* < 0.0001, n = 10; post hoc Tukey's multiple comparisons test, Gal80ts;Cas9;VT30559/+ vs Gal80ts;Cas9;VT30559/*blobby^gRNA* ****p* < 0.001, +/*blobby^gRNA* vs Gal80ts;Cas9;VT30559/*blobby^gRNA* ****p* < 0.001)., and (**G**) 3 h F(3,20) = 24.35, *p* < 0.0001, n = 6; one-way ANOVA followed by post hoc Tukey's multiple comparisons test, Gal80ts;Cas9;VT30559/+ vs Gal80ts;Cas9;VT30559/b*^lobbygRNA* ****p* < 0.001, +/*blobby^gRNA* vs Gal80ts;Cas9;VT30559/*blobby^gRNA* ****p* < 0.001. **F, H** Without induction (temperature control at 18 °C), flies showed normal memory scores. **F** F(3,18) = 1.003, *p* = 0.3902, n = 6 and (**H**) F(3,17) = 0.1574, *p* = 0.8558, *n* = 5. PI = performance Index. Graphs (**D–H**) show mean ± SEM. n: number of individual experiments performed. Source data are provided as a Source Data file.

key interaction surfaces necessary for binding to early scaffold components. In the absence of BRP, Blobby still localized near the residual AZ scaffolds (Fig. 2D). This indicates that while BRP is essential for Blobby's final incorporation into the AZ scaffold, it is likely not essential for the long-range axonal transport of Blobby. The association between BRP and Blobby may thus be particularly important for ensuring the directed, local transfer of axonally transported AZ precursors[47] at assembling AZs.

The absence of Blobby seems to alter the BRP scaffold nano-organization, as suggested by atypical BRP signals under STED microscopy (Fig. 4A) and the scarcity of typical T-bars at Blobby-deficient terminals (Fig. 5A, B). While a detailed molecular analysis is still pending, Blobby's structural features must be critical for its role in assembly. On the one hand, Blobby, similar to the BRP/ELKS family[48,49], contains predicted intrinsically disordered regions (IDRs) and coiled-coil (CC) domains. These features might play a role in the intricate process of AZ assembly, with IDRs promoting liquid-liquid phase separation (LLPS), a process meant to be critical for the ordered and efficient assembly of the AZ scaffold[1,50–54]. Indeed, recent findings from *C. elegans* suggest that a developmental liquid phase formed by the BRP homolog ELKS together with Syd-2/Liprin-α is important for the initial assembly of the synaptic AZ scaffold before maturation into a stable structure[55]. It is tempting to speculate, but certainly has to await appropriate experimental proof, that at nascent AZs in *Drosophila*, the IDRs of Blobby might play a role in maintaining BRP/ELKS-containing condensates in a liquid state during a crucial step of the AZ assembly process.

Blobby towards its N-term contains a TBC-type GTPase activating (GAP)-domain. TBC domains execute activity for Rab-family GTPases. Notably, Rab GTPases, have been implicated in the control of SV recruitment and their exocytic release[49,56], but also endocytic SV regeneration[56,57]. When compared to all human TBC-domain proteins in a dendrogram analysis, the Blobby TBC domain together sorts to the same node as TBC1D30, suggested to execute GAP activity for Rab3[58] based on recombinant protein and cell culture analysis. Rab3-GTP binds to SVs, where it is present in relatively high copy numbers[59]. At *Drosophila* larval NMJ AZs, *rab3* loss of function previously was shown to provoke an asymmetric distribution of BRP, with about half of the AZs presenting excess BRP, but the other half lacking a BRP AZ scaffold[60–62]. Similar to the *blobby* phenotype presenting ectopic blobs, *rab3* loss of function specifically affects the distribution of late AZ (BRP, RIM-BP, Unc13A) but not early (Liprin-α, Syd-1, Unc13B) components[8]. Analyzing to which degree dysregulation of Rab3-GTP might be part of the *blobby* phenotype warrants further analysis.

Phosphorylation has emerged as a key regulatory mechanism to dynamically regulate the interactions between scaffold proteins, thereby influencing their ability to undergo phase separation and assemble into functional AZ scaffolds. However, the specific phosphorylation events and their downstream effects on AZ assembly and function remain an active area of investigation[50,63–66]. A specific SRPK family protein kinase, SRPK79D, has been identified as a key player in the efficient transport and assembly of BRP[67,68]. Notably, a homologous SRPK in rodent hippocampal neurons has been implicated in both AZ assembly and functional maturation[65,69]. In *Drosophila*, SRPK79D phosphorylates a specific stretch within the N-terminal BRP IDR, a process crucial for efficient AZ incorporation[66]. This phosphorylation event prevents premature deposition of BRP prior to AZ incorporation, akin to observations in *blobby* mutants. Notably, aberrant "blobs" in the SRPK79D mutant also contain RIM-BP and Unc13A but not Syd-1 and Unc13B[66,70], essentially identical to the blobs within *blobby* mutants described in this study. Studying a potential functional interplay between Blobby and SRPK in AZ assembly could be a valuable subject for future analysis.

*Blobby* mutant NMJ terminals also exhibited significant physiological deficits. First, the quantal content, the number of SVs released per action potential across an NMJ terminal, was reduced. This reduction is likely attributed, at least to a large extent, to the decreased number of presynaptic AZs and the corresponding reduction in functional SV release sites, as demonstrated by mean-variance analysis.

In addition to the lack of functional release sites, we also found pronounced deficits in SV release probability, evidenced by atypical paired-pulse facilitation and further supported by the mean-variance analysis. Given that the physical transfer ("docking") of SVs to the AZ plasma membrane seems to remain largely intact (Fig. 5C–F), and that the proper alignment of SVs with $Ca^{2+}$ channels appears largely preserved (as indicated by our EGTA experiments, Fig. S7), our analysis suggests that inefficiencies in the biochemical preparation of SVs for fusion ("SV priming") might contribute to the observed release deficits in *blobby* mutants, alongside the reduction in release site numbers. While the *blobby* release deficit persisted also when using elevated $Ca^{2+}$ concentrations of 3 and 4.5 mM, its release probability deficits could be overcome at higher $Ca^{2+}$ concentrations (Fig. 7D). Although these $Ca^{2+}$ concentrations are likely unphysiologically high, the fact that $Ca^{2+}$-dependent regulation of the presynaptic release machinery can compensate for the putative priming deficits in Blobby-lacking AZ may provide clues for identifying the molecular cause of the deficits observed in *blobby* mutants.

While further studies are needed to clarify this issue, an interesting comparison arises from previous analysis of NMJs lacking the small GTPase Arl8[47,71], which is crucial for the effective transport of AZ precursor material. Although Arl8 lacking NMJs also exhibit reduced BRP levels and a lowered number of AZs, their paired-pulse ratios remain unchanged, indicating that SV release probability is unaffected here. This comparison also suggests that the absence of Blobby undermines AZ functionality, not just by reducing the amount of successfully integrated scaffold and release machinery. Future analyses should focus on determining which domains of Blobby, and its potential AZ interaction partners, are responsible for the observed deficits in SV release probability.

As mentioned above, at NMJ synapses, effective AZ clustering of Unc13A, the $Ca^{2+}$ channel α-subunit Cacophony (Cac), and RIM-BP is BRP-dependent[18,37,72]. In *blobby* mutants, our STED analysis of these critical components for evoked SV release did not reveal major deficits yet. However, nanoscale deficits, potentially detectable through single-molecule analysis[16], might expose more subtle issues. BRP conformational state changes in response to Blobby could affect priming efficacy, possibly through interactions with Unc13A or RIM-BP. It appears equally possible that changes in Rab protein activity contribute to the presumed priming inefficiencies.

The identification of Blobby signifies the emergence of a specialized assembly factor, suggesting the evolution of regulatory mechanisms tailored to the demands of the AZ assembly process. Notably, the absence of Blobby resulted in fewer adult flies hatching (Fig. 8A), accompanied by defective locomotion (Fig. 8B). These data suggest that the developmental deficits of *blobby* mutants in the nervous system, exemplified in our study by analyzing larval NMJs, cannot be fully compensated for. Notably, however, we observed Blobby across synaptic regions in the adult, post-developmental *Drosophila* brain. However, a post-eclosion knockdown of *blobby* ("bypassing" the developmental requirement) specifically in the mushroom body Kenyon cells completely abolished olfactory memory consolidation. This phenotype is similar to the effects observed with BRP knockdown under the same conditions[46,73]. Thus, Blobby is likely a regulatory component of the BRP-orchestrated AZ scaffold, also playing a crucial role in AZ remodeling processes essential for memory consolidation at a subset of MB synapses.

Taken together, our analysis identifies an additional layer of AZ regulation critical for developmental AZ assembly but also for AZ-mediated plasticity controlling behavior.

## Methods

### Fly husbandry, stocks and handling

Fly strains were reared under standard laboratory conditions and raised on semi-defined conventional cornmeal-agar medium (Bloomington recipe) with 60-70 % humidity at 25 °C, or 18 °C for adult aversive associative olfactory conditioning. For all experiments, both male and female third instar larvae or adult flies were used. For proteomics/WB, confocal and STED stainings, electrophysiology the following genotypes were used: *w*[1118] (ctrl.), *blobby*[Null], *blobby*[GFP], *blobby-KDRT-4xSTOP-KDRT-ALFA*, *ok6-Gal4* (Bloomington 64199) driver, *mef2-gal4* (Bloomington 600192) driver and a KD recombinase (BL# 55791). The *sfCac*[GFP] (*cac*[GFP]) fly *line*[74] were generously provided by Kate M. O'Connor-Giles (Brown University). For adult behavior, *Gal80*[ts]*,Cas9/+;VT30559/+*, *Gal80ts,Cas9/blobby*[gRNA]*;VT30559/+* were used.

### Generation of transgenic flies

The following flies were generated in cooperation with Well Genetics Inc. (Taipei City, Taiwan) via CRISPR/Cas9-mediated genome editing by homology-dependent repair (HDR) using guide RNAs and a dsDNA plasmid donor according to Kondo and Ueda[75]: *blobby*[Null], *blobby*[GFP], and *blobby*[gRNA].

The *blobby*[Null] allele was produced by two consecutive deletion steps, consequently introducing a 4724-bp deletion (deletion region: +13,885 nt to +18,608 from ATG of CG42795-RA) followed by a second deletion of 5,000-bp (deleted -4,997 nt to +3 nt from Stop Codon of CG42795-RC/F).

The *blobby*[GFP] allele was produced by knock-in the eGFP tag right after S772 (based on blobby-RG isoform). The eGFP sequence is flanked by an 8 aa-linker (AGCTGTCTCTTATACACATCTGGC) upstream and a 12aa-linker (GGCGCGCCCGGGCAGATGTGTATAAGAGACAGAGGC) downstream of eGFP.

The conditional *blobby-KDRT-STOP-KDRT-ALFA* line was created by inserting the *KDRT-STOP-KDRT-ALFA* directly after the amino acid S772 of the CG42795-PG isoform. For the analysis of presynaptic and postsynaptic KDRT expression, an *ok6-Gal4* (Bloomington 64199) driver or *mef2-gal4* (Bloomington 600192) driver and a KD recombinase (BL# 55791) were combined with the conditional *blobby-KDRT-STOP-KDRT-ALFA*. The 3rd instar larvae with the correct genotype were collected and dissected.

The *blobby*[gRNA] allele for in-vivo TRiP-CRISPR gene knockout was designed to target all isoforms of *blobby* (the large CG42795 isoforms *RA, RC, RE, RF,* and *RG* are targeted by the gRNA in the second exon; the small *RB* isoform is targeted in the first exon). The gRNA GTGGCTGCATGCCATGAAAC[TGG] (CRISPR target site [PAM]) was generated by gene synthesis using the following primers: sense-oligo 5′- GTCG TGGCTGCATGCCATGAAAC; antisense-oligo 5′- AAAC GTT TCATGGCATGCAGCCA.

### Eclosion rate

The eclosion frequency of the adult flies (2–3 days old) was determined as a measure of viability. Briefly the eclosion rate was recorded from seven independent crosses. Each cross was set up with 15 *blobby*[Null]/TM6B,Tb virgins and 10 *blobby*[Null]/TM6B,Tb male flies. The entire F1 generation was examined and evaluated based on the following criteria: presence (heterozygous) or absence (homozygous for *blobby*[Null]) of the balancer chromosome. The total number of eclosed flies was counted and expressed as a percentage as a function of the survival rate. Statistical significance was assessed using the chi-squared test in GraphPad Prism.

### Generation of Blobby specific antibodies

Blobby[C-Term] specific antibody. The polyclonal antibody was raised in rabbits. Immunization of the animals was performed using a His -tagged fusion protein. The coding sequence corresponding to aa 2090 - 2422 of Blobby (CG42795, based on isoform G) was MTEK KETIKDSSSKELPEKMVINSTDVGPMDPNGKTVVLLMDNEHRASKVRRLT RANTEELEDLFQALEKQLNDRNLVKSEDGRLIRVDPKPSAEQVEQTQAISD LTKEIEDFTSAKPEEENPKEAAKEDKPEPEEPEDFDWGPNTVKHHLKRKTV YLPSTKELESRFRSLERQIKLLEDVEKIDVEQRLNEIERKIKLQYSLSHEKDLN KYLELCEGKGLDDDEPVPVETPTKEAEITTARDRSRSPGRKALATKSPYTS PSRKATIKTPHTSPTRKPIIKSPYTSPSRKSAKSPYTSPSRNRQRSPSPTRSP ERKSKKSPYTSPARRKPHP.

**The PCR was performed using the following primer.** Blobby C-term Fwd, 5′ GATCCCATGGGGGCCAAACGGAAAAACAGTGGTGCTG and Blobby C-term Rev, 5 GATCGCGGCCGCTTACATTCGCTTGGCCAAGT TCTCCTTAG.

The PCR product was subcloned into pETM-11 (HIS-tag vector). The expression and purification of the target protein were performed in *Escherichia coli* under native conditions. Immunization and affinity purification of the AB containing serum was performed at BioGENES company (Berlin), using the same HIS-tagged fusion protein used for immunization. Antibody specificity was tested in western blot (Fig. 1B) and staining of *blobby*[Null] mutants (Fig. 1E).

The Blobby[ex8b] specific antibody was raised in rabbits. Immunization of the animals was performed using a His-MBP-tagged fusion protein. The coding sequence corresponding to amino acids 1019-1487 of Blobby (CG42795, based on isoform G) is EIEERYQALERRISQ DQPSGDRQAKYIPSTAALEERFNTLEKQLSAEKQRKELSEMEAEYPIKSER-IPSTADLESRFNSLTKQMSSSESSSKTPIDLKDEDRPSGSSSKNQKDSEKT SKLHKSEEPESNTKETTGETEASDSNDSKIGEKETEQPRIKKLPSTAELEDR FNALERKMSVQKSSPSKNKKEPPDEEESKSTKEPEEPEESEKANEKTSGRQT PIAKKDSKDSDQKKSETKENQSPTKNQDEKVKVKSPKSEEMIEKETSSNPK EDSHESEAATNKKVEGNRELSSEKGDHKIKEKSEEAPGKAGKETAETKNAN VKDSSKKGDSQKNEAAKTSVSQTESDLKPSSKENSTSKDAEQEKTPRKSP PSTEELEKRFNALEKQMSTTNLETTKEPDQTKPATKSQSTSAEVKTQKSM KSFDDKIKEVNVAIEKEQSRVEVEVNAEKKRKNVEEAPKNKEGDSQ.

The insert was amplified by PCR. The forward primer sequence was 5′ CCGCCATGGAAATCGAGGAACGGTATCAG 3′ and the reverse primer sequence was 5′ AAGCGGCCGCCTACTGAGAATCCCCTTCTTT GTTTTTTGGGG 3′.

The PCR product was subcloned to pETM-42 vector containing an N-terminal His-MBP tag. The expression and purification of the target protein were performed in *Escherichia coli* under native conditions. Immunization and affinity purification of the AB containing serum was performed at BioGENES company (Berlin), using the same His-MBP-tagged fusion protein used for immunization. Antibody specificity was tested in western blots (Fig. 1B) and staining of *blobby*[Null] mutants (Fig. 8C).

### Generation of the Unc13A specific antibody

The polyclonal antibody was raised in guinea pigs. Immunization of the animals was performed using a His -tagged fusion protein. The coding sequence corresponding to aa 1337 - 1632 of Unc13A was:

NILPIGPQATGKKLPTVNGKSALLIKQMPTEVYDDESDTDELDVSPST GKVPSYSIYSEQEDYYMDLQQTTPSIQPNGFYEQVNNGYDYREDYFNEED EYKYLEQQREQEEHNQPKNKKYLKQAKISKIQPPSLDFIDVGQDDDFIYD NYHSEDDSGNYLEGSSSGSVGPIEGSIIKVDSNIEASFASLNKKSDSFTPTND SLQKHDTVIGESTTKLTRLRTEKMCPDVDEEDENLSDHVSDLTDLSKLISQ KKKTLLRGETEEVVGGHMQVLRQTEITARQRWHWAYNKIIMQLN.

**The PCR was performed using the following primer.** Unc13A_AK5 Fwd, 5′ GATCCCATGGGGAACATACTCCCGATTGGCCCGCAGGC and Unc13A_AK5 Rev, 5 GATCGCGGCCGCTTAATTAAGCTGCATGATTATT TTATTG.

The PCR product was subcloned to pETM-11 (HIS-tag vector). The expression and purification of the target protein were performed in *Escherichia coli* under native conditions. The HIS-tagged fusion protein was used for the affinity purification of the AB-containing serum, which was obtained from Selbaq. 19GP02 Antibody specificity was

tested in western blot and staining of in Unc13A mutants (data not shown).

## Western Blot analysis of adult brain protein extract

Western Blot analysis was performed as previously described in Huang et al.[27], with some modifications. In brief, brains of 10 days old animals were dissected in HL3, homogenized in lysis buffer (0.5% Triton X-100, 2% SDS, 1× protease inhibitor, 1× sample buffer in PBS) followed by full-speed centrifugation at 18 °C. One brain´s supernatant was loaded to SDS-PAGE and immunoblotted according to standard protocols. The following primary antibodies were used: rabbit anti-Blobby[C-Term] (1:500), rabbit anti-Blobby[ex8b] (1:500), mouse anti-Tubulin (Sigma T9026, 1:100,000). Uncropped and unprocessed scans of Western blots (from Fig. 1B, C) are provided in the Source Data file.

## Isolation and purification of *Drosophila* synaptosomes

The procedure involves decapitation of adult flies (sieving, ~6000 heads from 10–20 days old flies), pulverization, homogenization (320 mM sucrose), 4 mM HEPES, protease inhibitors (complete, 11873580001, Roche) and differential centrifugation (from low speed to higher speed: 1,000–15,000 g) of fly heads, which allows subsequent isolation and enrichment of presynaptic and postsynaptic components[28].

## BRP Co-immunoprecipitation

Co-Immunoprecipitation experiment of Bruchpilot was performed with crude synaptosomes resuspended in homogenization buffer containing 320 mM Sucrose, 4 mM HEPES and a protease inhibitor cocktail, pH 7.2 as described in Depner et al.[28]. Approximately 6000 fly heads were collected per replicate, and synaptosomes were purified via differential centrifugation (see Depner et al.[28]). Approximately 20 µg of guinea pig BRP[last200] antibody was coupled to 50 µl Protein A–coated agarose beads. For a negative control guinea pig-IgGs were coupled to beads. To avoid unspecific bounds of proteins to beads, synaptosome suspension was precleared by rotating for 1 h at 4 °C on naked beads. Afterwards, bead-antibody and bead-IgG complexes were incubated with solubilized and precleared synaptosome membrane preparations (P2) overnight at 4 °C. After four washing steps (20 min each) with IP buffer (containing 20 mM Hepes, pH 7.4, 200 mM NaCl, 2 mM MgCl$_2$, 1 mM EGTA and 1% Triton X-100), antibody antigen complexes were eluted with 60 µl 2×denaturing protein sample buffer and subjected for mass-spectrometry as well as for WB analysis.

## Proteolytic digestion of BRP-IP eluate

BRP-IP eluate and IgG eluate (4 biological replicates each) were reduced (5 mM DTT at 37 °C for 60 min), alkylated (40 mM CAA at RT for 30 min, dark) and loaded on NuPAGE 4 – 12% Bis-Tris SDS-PAGE (see Source Data). Subsequently, each lane was divided in three separate slices and independently subjected to an in-gel trypsin digestion procedure (enzyme:protein ratio of 1:20 (wt/wt) at 37 °C overnight). Each slice was submitted for mass spectrometry analysis, resulting in 24 acquisitions and merged during data analysis.

## LC-MS/MS analysis

Equal volumes corresponding to 1 µg peptide were loaded on a Thermo Scientific Dionex UltiMate 3000 system connected to a PepMap C-18 trap-column (0.075 × 50 mm, 3 µm particle size, 100 Å pore size; Thermo Fisher Scientific) followed by an in-house-packed C18 column (column material: Poroshell 120 EC-C18, 2.7 µm; Agilent Technologies). Peptides were separated at 250 nL/min flow rate over a 120 min gradient of increasing acetonitrile concentration and sprayed into an Orbitrap Fusion Lumos (instrument control software 3.1). The MS1 scans were performed in the Orbitrap in positive mode with the following settings: 120,000 resolution, scan range 375 – 1,500 m/z, the

50 ms max. injection time, AGC target 400,000. Only precursors with a charge state of 2–4 were subjected to fragmentation and then dynamically excluded for 60 s. The MS2 scans were acquired in the ion trap with the following settings: scan rate rapid, 35 ms max. injection time, first mass 110 m/z, isolation window 1.6 m/z, 30% NCE, AGC target 10,000. A 1 s cycle time was set between master scans.

## Database search and label-free quantification analysis

Raw data were searched using MaxQuant version 1.6.2.6 using default settings[76]. The number of missed tryptic cleavages allowed was set to 2, label-free quantification was enabled, and the match-between-runs option was disabled. The search was performed using the UniprotKB database of Drosophila melanogaster downloaded on May 2020 containing 42,678 entries. Both, PSM and protein FDR have been set to 1%. Using the Perseus software[77], LFQ values were log2 transformed to achieve a normal data distribution. Proteins identified in at least three (out of four) replicates were considered for statistical analysis. Proteins that were detected and quantified in only one replicate were excluded. Missing data were imputed by values from a normal distribution (width 0.3 standard deviations; down shift 1.8). For statistical protein enrichment analysis in the BRP-IP, a two-sided *t*-test between BRP-IP and negative IgG control was used to calculate a *p*-value. Presented fold changes have been calculated as the difference of mean values from log2 transformed intensities from BRP-IP and the IgG control. Microsoft Excel was used to create the Volcano plot from quadruplicates of coprecipitated protein levels from the BRP-IP compared with the IgG control. The *x*-axis represents the log2 fold-change, indicating the magnitude of change, and the y axis is –log10 of the *p*-value showing statistical significance.

## Immunostaining of *Drosophila* larval tissue

Larval filets were dissected and stained as previously reported in Owald et al.[78]. Briefly, third instar larvae were dissected in ice-cold hemolymph-like saline (HL3; composition in mM: NaCl 70, KCl 5, MgCl$_2$ 20, NaHCO3 10, trehalose 5, sucrose 115, HEPES 5, pH adjusted to 7.2) and fixed in either 4% paraformaldehyde in PBS (pH 7.2) for 10 min for all antibodies or in ice-cold methanol for 5 min for Unc13A antibody. Afterward, the larval filets were washed in PBS containing 0.05% Triton X-100 (PBST) and blocked for 1 h in 10% ROTI Block (Carl Roth). The primary antibody incubation was performed at 4 °C overnight. Secondary antibody incubation was carried out for 3hrs at room temperature. Immunocytochemistry was equal for both conventional confocal and STED microscopy. The following primary antibodies were used: mouse anti-Bruchpilot[Nc82]/ BRP[Cterm] (1:100, DSHB, catalog #Nc82; RRID:AB_2314866), rabbit-anti Blobby[C-term] (1:300, this manuscript), anti-rabbit Blobby[ex8b] (1:500, this manuscript); guinea pig-anti Unc13A (1:300, this manuscript); rabbit-anti Unc13B; rabbit-anti GluRIID (1:500, Qin et al.); rabbit anti-RIM-BP[18]; rabbit anti-Syd-1[78]; FluoTag-X2 anti-ALFA AbberiorStar635P (N1502-Ab635P-L). The secondary antibodies for standard immunostaining were used at the following concentrations: goat anti-HRP-Cy5 (1:250, Jackson ImmunoResearch); goat anti-rabbit-Cy3 (1:500, Jackson ImmunoResearch 111-165-144); goat anti-mouse or anti-rabbit Cy3 (1:500, abcam, ab97035/ ab6939); goat anti-mouse or anti-guinea pig or anti-rabbit Alexa Fluor 488 (1:500, Life Technologies A11001/ A11073/A11008). For confocal microscopy larvae were mounted in Vectashield (Vector Labs). For STED microscopy the larvae filets were embedded in Prolong Gold Antifade (Invitrogen).

Secondary antibodies for gSTED microscopy were used in the following concentrations: Alexa Fluor594-coupled goat anti-rabbit (Invitrogen A32754, 1:300), STARRED FluoTag X2-coupled goat anti-mouse (Abberior STRED-1001-500UG, 1:300); anti-GFP STARRED Fluo Tag X4 (1:300 for STED, NanoTag N0304-AbRED-L); goat anti-mouse ATTO490LS (1:50 for STED, Hypermol Cat.#:2109-1MG).

## Immunostaining of *Drosophila* adult brains

Adult Brains were dissected in ice-cold hemolymph-like saline (HL3; composition in mM: NaCl 70, KCl 5, MgCl2 20, NaHCO3 10, trehalose 5, sucrose 115, HEPES 5, pH adjusted to 7.2) solution and immediately fixed in 4% paraformaldehyde (pH = ~7.3) for 30 min at room temperature. After fixing, brains were washed in 0.7% PBST (PBS with 0.7% Triton X-100, v/v) for 3 or 4 times for a total of 1 h and blocked in 0.7% PBST with 10% normal goat serum (v/v) for at least 2 h at room temperature. Primary antibodies were diluted in 0.7% PBST with 5% NGS for primary antibody incubation at 4°C under stirring for 48 h. Afterwards, brains were washed again in 0.7% PBST for at least 4 times and then incubated with secondary antibodies diluted in 0.7% PBST with 5% NGS in darkness overnight. Finally, after secondary antibody incubation, brains were washed for at least 4 times and samples were embedded in Prolong Gold Antifade (Invitrogen) and stored for 24 hr at room temperature followed by 48 hr at 4 °C. For STED microscopy following primary antibodies were used: mouse anti-BRP[Nc82] (DSHB, 1:10), anti-Blobby[Ex8b] (1:100) and guinea pig anti anti-Drep2 (1:200). ATTO490 LS-coupled goat anti-rabbit (Hypermol 2309, 1:50), Alexa Fluor594-coupled goat anti-guinea pig (Invitrogen 11076, 1:250), STARRED-coupled goat anti-mouse (Abberior STRED-1001-500UG, 1:250) were used for secondary antibody incubation.

## Confocal, STED and live microscopy

Time-gSTED and corresponding confocal laser scanning microscopy were performed using an Abberior Instruments Expert Line STED setup equipped with an inverted IX83 microscope (Olympus), two pulsed STED lasers for depletion at 775 nm (0.98 ns pulse duration, up to 80 MHz repetition rate) and at 595 nm (0.52 ns pulse duration, 40 MHz repetition rate) and pulsed excitation lasers (at 488 nm, 561 nm, and 640 nm), operated by Imspector software (16.3.15507, Abberior Instruments, Germany). The dyes STARRED, Alexa Fluor594, and ATTO490 LS were depleted with a pulsed STED laser at 775 nm. Time gating was set at 750 ps. Fluorescence signals were detected sequentially by avalanche photodiode detectors at appropriate spectral regions. 2D confocal and corresponding gSTED Images were acquired sequentially with a 100x, 1.40 NA oil-immersion objective, with a pixel dwell time of 2 µs and 10x or 30x lines accumulation, respectively, at 16 bit sampling and a field of view of 10 µm × 10 µm. Lateral pixel size was set to 20 nm. Within each experiment, samples belonging to the same experimental group were acquired with equal settings. Raw triple channel gSTED images were processed for Richardson–Lucy deconvolution using the Imspector software (16.3.15507, Abberior Instruments, Germany). The point spread function was automatically computed with a 2D Lorentz function having a full-width half-maximum of 40 nm, based on measurements with 40 nm Crimson beads. Default deconvolution settings were applied.

Confocal microscopy was performed with a Leica SP8 microscope (Leica Microsystems). Images of fixed and live samples were acquired at room temperature. Confocal imaging was done using a z-step of 0.3 µm for fixed NMJs and 0.25 µm for live imaging. 63× 1.4 NA oil immersion objective was used for NMJ confocal imaging. All confocal images were acquired using the LCS AF software (Leica Microsystems). Images were taken from third instar larval NMJs (segments A2 – A4). Images for figures were processed with ImageJ software to enhance brightness using the brightness/contrast function. If necessary, images were smoothened (0.5 pixel Sigma radius) using the Gauss blur function. Confocal stacks were processed with ImageJ software (http://rsbweb.nih.gov/ij/). Quantifications of AZ spot number, density and size (scored via BRP) were performed as described in Owald, et al. and Andlauer, Sigrist 2012[78–81]. Live imaging, and particle analysis on 0 hr data, was performed as described in Ramesh et al.[81]. The resulting data was then binned based on AZ spot area (0.1µm² bins) for each NMJ to plot intensities of BRP and Blobby and Blobby/BRP intensity ratios.

To quantify the ectopic BRP positive material, ROIs for each identified BRP-positive spot was generated by using ImageJ software. Following, the ectopic BRP material was calculated by forming the ratio of BRP[Nc82] intensity signal divided by GluIID intensity within each identified ROI. Next, the standard deviation of the ratios were determined. The cutoff was calculated by the sum of 3x standard deviation and mean of ratios. Each ratio higher than the calculated cutoff was considered as ectopic material. BRP positive ectopic material bigger than 0.3 µm are considered as 'blobs'.

## gSTED peak-to-peak distance analysis

Deconvolved 8-bit gSTED images were used for quantification of peak-to-peak distances by line profile measurements. Line profile measurements of distances between spots were performed in ImageJ (version 1.52p, NIH). Well-defined side view synapses were manually traced with the line profile tool (thickness 9 pixels/180 nm) and peak intensities across the line were retrieved using the ImageJ Macro (Macro_plot_lineprofile_multicolor from Kees Straatman, University of Leicester, Leicester, UK). Intensity values from individual synapses were exported to Excel. Local maxima were calculated with the SciPy "argrelmax" function, as described in Brockmann[79], in order to obtain peak intensities for different image channels and peak-to-peak distances. Only highest maxima were selected[80]. Values were then averaged per animal.

## Analysis of nearest neighbor inter-CacGFP distances

The XY coordinates of presynaptic sfCac[GFP] clusters within 1 µm radius from a given sfCac[GFP] cluster were determined on 8-bit deconvolved gSTED images by their peak locations detected with the Find Maxima plugin (ImageJ, version 1.52p, NIH), similarly to Fukaya et al.[82], with prominence set to 10. Then the Euclidean distances between the first closest sfCac[GFP] neighbors were retrieved. Only one sfCac[GFP] spot per bouton within an image was considered.

## Perimeter: Segmentation and characterization of the AZ nanostructure

AZs were segmented using a custom ImageJ script (available at https://github.com/ngimber/BruchpilotSegmentation) based on the BRP signals from gSTED images. For AZ identification, images were processed by applying a Fourier bandpass filter for medium-to-large structures (0.1–2 µm). The AZs were then identified using the built-in 'MaxEntropy' auto-thresholding algorithm and 'watershed' algorithm from ImageJ[8]. Small clusters, likely representing immature AZs, were identified by applying a Fourier bandpass filter for small structures (0–0.06 µm) to the original images and applying the 'Minimal' threshold algorithm[8] from ImageJ. These structures were excluded from the AZ quantification. The perimeters of identified AZs were measured, and the results were plotted as Mean ± SEM from 22-26 boutons (4-5 animals).

## Electron microscopy

Conventional embedding of larval muscles was performed as previously described in Matkovic et al.[40] In brief, dissected third instar larvae were fixed with PFA (10 min; 4% PFA and 0.5% glutaraldehyde in 0.1 M PBS) and glutaraldehyde (60 min; 2% glutaraldehyde in 0.1 M sodium cacodylate), washed in sodium cacodylate buffer, and post-fixed with 1% osmium tetroxide and 0.8% KFeCn in 0.1 M sodium cacodylate buffer (1 h on ice). After washing with sodium cacodylate buffer and distilled water, the samples were stained with 1% uranyl acetate in distilled water. Samples were dehydrated and infiltrated in EPON resin. Subsequently, muscles 6/7 of the abdominal segment A2/3 were cut out. Collected in an embedding mold, the blocks were polymerized and cut into thin sections of 60 nm using a Leica EM UC7 ultramicrotome equipped with a 3 mm diamond knife (Diatome). Postcontrasted sections were imaged at 11000× and

21000x nominal magnification on a Tecnai Spirit transmission electron microscope (FEI).

## High pressure freeze electron microscopy (HPF)

High pressure freeze electron microscopy (HPF) embedding was performed as described previously[40] with some modifications. In brief, three *Drosophila* late second/early third instar larvae were dissected (not fixed), placed in an aluminum specimen carrier of 200-µm depth (HPF carrier type A, Leica Microsystems GmbH, Wetzlar, Germany), filled with 15% Ficoll400 (Carl Roth GmbH + Co. KG, Karlsruhe, Germany), and covered with a lid (HPF carrier type B, Leica Microsystem GmbH, Wetzlar, Germany). Larvae filets were frozen under high pressure immediately in the HPF Leica EM ICE system (Leica Microsystem GmbH, Wetzlar, Germany) and stored in liquid nitrogen. Cryosubstitution was performed in an AFS (Leica) in anhydrous acetone with 1% EMD Millipore water, 1% glutaraldehyde, and 1% osmium tetroxide. From −90 °C for 10 h the temperature was slowly (5 °C/h) increased to −20 °C, the samples were incubated for additional 12 h before being warmed (10 °C/h) to 20 °C. Following the samples were washed with acetone and incubated with 0.1% uranyl acetate (dissolved in anhydrous acetone) for 1 h at room temperature (RT). After washing, the samples were infiltrated with the plastic resin Epon in increasing concentrations. The first incubation step in 30% Epon/70% acetone for 4 h was followed by 70% Epon/30%acetone overnight. The samples were incubated twice in 100% Epon for 2 h before being embedded.

Ultrathin sections of 60 nm thickness were cut using a Leica EM UC7 ultramicrotome equipped with a 3,5 mm diamond knife (Diatome). Micrographs were acquired on a postcontrasted sections at 11000× and 21000x nominal magnification on a Tecnai Spirit transmission electron microscope (FEI).

## Quantification of HPF EM images

High-pressure frozen (HPF) electron micrographs (4096 ×4096 pixels) of neuromuscular junction (NMJ) boutons at 11,000x and 21,000x magnification, with pixel sizes of 0.98 nm and 0.512 nm respectively, were manually analyzed using ImageJ software (National Institutes of Health). Only type Ib boutons, characterized by their surrounding dense subsynaptic reticulum (SSR), were included in the analysis. Active zones were identified by parallel alignment of presynaptic and postsynaptic membranes.

Measured parameters included bouton area, bouton perimeter, and AZ length, all of which were measured using the freehand line tool in ImageJ. For the quantification of vesicles distribution relative to the AZ membrane, a rectangular region of interest (ROI) measuring 400 nm x 500 nm, was defined, centered on an individual, straight AZ membrane. Only vesicles located within 450 nm were considered. The orthogonal distance between the center of each vesicle within the ROI and the inner leaflet of the plasma membrane (PM) was manually measured using the straight-line tool. Vesicle diameters were measured by delineating their profiles with the oval tool.

For the quantification of docked vesicles, the shortest distance between the outer leaflet of the SV membrane and the inner leaflet of the AZ plasma membrane was manually measured using the straight-line tool in ImageJ. Vesicles attached to the membrane (0–5 nm distance) and those close to the AZ membrane (5–10 nm distance) were categorized accordingly.

## Electrophysiology

Two-electrode voltage clamp (TEVC) recordings were performed essentially as previously reported (Petzoldt et al.). They comprised spontaneous recordings (miniature excitatory junction currents: mEJCs, 90 s), single evoked (evoked excitatory junction currents: eEJCs, 20 repetitions at 0.2 Hz) and high-frequency recordings (paired-pulse 10 ms or 30 ms interstimulus interval, PP10 or PP30, 10

repetitions at 0.2 Hz; 60 pulses at 100 Hz for cumulative quantal content computation) as well as mean variance analysis (5× eEJC protocol at different c[Ca$^{2+}$]). All experiments were performed on third instar larvae raised at 25 °C. The dissection and recording medium was extracellular HL3 solution. Dissection was performed in Ca$^{2+}$+-free HL3 medium at room temperature, while mEJC, eEJC, and high-frequency recordings were performed in 1.5 mM Ca$^{2+}$+HL3 at room temperature. Data for mean-variance analysis were recorded at 0.325, 0.75, 1.5, 3, and 4.5 mM Ca$^{2+}$ by starting with a bath volume of 2 ml 0.325 mM Ca$^{2+}$ + HL3, consecutively removing 1 ml of the former bath solution and adding 1 ml of 1.175, 2.25, 4.5 or 6 mM Ca$^{2+}$+HL3, respectively, while mixing carefully with a pipette and giving 1 min of acclimation period before the next measurement. For all physiological recordings, intracellular electrodes with a resistance of 20−35 MΩ (filled with 3 M KCl) were placed at muscle 6 of the abdominal segment A2/A3. The data acquired were low-pass filtered at 1 kHz and sampled at 10 kHz. The command potential for mEJC recordings was −80 mV, and −60 mV for all other recordings. Only cells with an initial membrane potential between −50 and −70 mV and input resistances of ~4 MΩ were used for further analysis. The eEJC and paired-pulse traces were analyzed for standard parameters (amplitude, rise time, decay, charge flow, paired-pulse [PP]-ratio) by using a semiautomatic custom written Matlab script (Mathworks, version R2009a). The 100-Hz trains were analyzed for amplitudes by using another semiautomatic custom-written Matlab script that calculates eEJC amplitudes by measuring peak to baseline directly before the onset of the response. The quantal content of each response was calculated by dividing the amplitude by the mean quantal size of the respective genotype. Release-ready vesicles (y intercept) and refilling rate (slope) were determined by back extrapolation of the last 300 ms of cumulative quantal contents. Mean variance analysis was basically performed as described previously[10]. In short, the amplitudes of 7 repetition traces per c[Ca$^{2+}$] were averaged (first ten without 1−3 to reduce possible effects of "super-priming") and plotted against mean variance of the amplitudes (SD2) to obtain the mean versus variance plot and parabolic fits. Second-order polynomial fits (SD2 = q × Ī − Ī 2 /N) were performed per cell where q is the quantal size, Ī is the mean current amplitude, and N is the number of release sites. Vesicular release probability was calculated by PVR = Ī /(N × q) per cell. The parabolas in Fig. 7B represent fits to the mean values of a full dataset per genotype. The Hill plot represents the amplitude per cell at the respective c[Ca$^{2+}$] normalized to the maximal amplitude of the same cell at 4.5 mM c[Ca$^{2+}$], averaged per genotype. Stimulation artifacts in eEJC and paired-pulse recordings were removed for clarity. The mEJC recordings were analyzed with pClamp 10 software (Molecular Devices). GraphPad Prism v5.01 (GraphPad Software, Inc.) was used for all fitting procedures. Data were analyzed using GraphPad Prism v8.4.2. Data distribution is normal following the D'Agostino and Pearson omnibus normality test. If data had a normal distribution, an unpaired, two-sided *t*-test was applied for comparison of two conditions or a one-way ANOVA using Tukey post-test if more than two conditions were compared. If data did have a normal distribution, the nonparametric, two-tailed Mann−Whitney test was applied for comparison of two conditions or the nonparametric Kruskal−Wallis test with the Dunn's post-test if more than two conditions were compared. For standard TEVC analysis, n = 1 cell, and one or two cells from four to six animals were analyzed.

## Electrophysiology under EGTA-AM

For EGTA-AM experiments, fully dissected larvae were incubated for 30 min at room temperature in HL3 medium containing 0.1 mM EGTA-AM (Calbiochem, 50 mM stock solution in DMSO) and 0.1 mM Pluronic F-127 (Molecular Probes, 20% solution in DMSO). As control the same amount of DMSO and Pluronic was dissolved in HL3. After incubation the preparation was rinsed three times with HL3. Data was analyzed with a custom-written Python 3.10 script utilizing the pyABF package

(Harden, SW (2022). pyABF 2.3.5. [Online]. Available: https://pypi.org/project/pyabf).

## Behavior: olfactory associative aversive conditioning

Flies were trained using the classical olfactory aversive conditioning protocols as described by Tully & Quinn[83]. Training and testing were performed in climate-controlled boxes at 25 °C in 80 % humidity under dim red light. At 2–3 days old, flies were transferred to fresh food vials and either put at 29 °C for gRNA induction or stayed at 18 °C for the non-induced controls. Conditioning was performed on groups of around 40–50 flies with 3-octanol (around 95 % purity; Sigma-Aldrich) and 4-methylcyclohexanol (99 % purity; Sigma-Aldrich). Odors were diluted at 1:100 in paraffin oil and presented in 14 mm cups. A current of 120 AC was used as a behavioral reinforcer. Memory conditioning and tests were performed with a T-maze apparatus[83]. n a single-cycle training paradigm, groups of flies were exposed to one odor (CS +) paired with an electric shock (US) 12 times over the course of 1 min. Following a 1-min period of pure airflow, a second odor (CS-) was presented without the shock for another minute. During the test phase, flies were given 1 min to choose between two arms, each containing a distinct odor. A performance index (PI) was calculated as the difference between the number of flies in each arm, divided by the total number of flies in both arms. The average of two reciprocal experiments yielded the final PI. PI values ranged from 0 to 1, where a value of 0 indicates no learning (a 50:50 distribution of flies), and a value of 1 represents complete learning (all flies avoided the conditioned odor).

For short-term memory (STM) testing, flies were evaluated immediately after conditioning. For middle-term memory (MTM), flies were transferred to small tubes without food and tested after either 1 h (MTM 1 h) or 3 h (MTM 3 h). To assess long-term memory (LTM), flies underwent three training cycles spaced by 15 min rest intervals, then were kept in standard food vials at 29 °C for 24 h before memory testing.

For olfactory acuity and shock reactivity tests, ~50 flies were placed in a choice situation between one odor and clean air (for olfactory acuity), or between electric shocks and no-shocks (for shock reactivity), each for 1 min.

## Startleinduced negative geotaxis (SING)

SING tests were conducted following the method described by ref. 84, with a few modifications outlined here. For each genotype, ~100 flies (age 5–6 days) were tested, consisting of ~55 adult males and ~45 adult females. These flies were divided into groups of 10–15, independent of sex, and each group was tested separately. The test was carried out in a vertical glass cylinder (23 cm in length, 2.5 cm in diameter). At the beginning of the experiment, the flies were placed in the cylinder and allowed to habituate for 30 min. The test began by gently tapping the cylinder downwards, prompting the flies to respond by climbing upwards. After 1 min, the flies that reached the top of the column (above 20 cm) and those remaining at the bottom (below 4 cm) were counted. Each group of flies was tested three times at 15 min intervals. The performance index (PI) for each column was calculated as $\frac{1}{2}[1 + (n\_top - n\_bot) / n\_tot]$, where n_tot is the total number of flies in the column, n_top is the number of flies at the top, and n_bot is the number of flies at the bottom after 1 min. This assay was performed on 5–6 day-old flies. Source data are provided as a Source Data file.

## Statistics and reproducibility

Data were analyzed using Prism (Version 7 & 8, GraphPad Software). Per default unpaired Student's *T*-test was performed to compare the means of two groups unless the data were either non-normally distributed (as assessed by D'Agostino-Pearson omnibus normality test) or if variances were unequal (assessed by *F*-test) in which case they were compared by a Mann–Whitney *U* Test (differences between 2

groups based on average ranks) or two samples Kolmogorov-Smirnov test (differences in the shape of two distributions between two groups). For comparison of more than two groups, one-way analysis of variance (ANOVA) tests were used, followed by a Tukey's multiple comparison test. For immunostaining, all genotypes were prepared in one session, stained in one cup and analyzed in an unbiased manner. For electrophysiological recordings, genotypes were measured in an alternating fashion on the same day and strictly analyzed in an unbiased manner. *p*-values and n values are given in the figure legends. Means are annotated ± s.e.m. Asterisks are used to denote significance: *$p < 0.05$; **$p < 0.01$; ***$p < 0.001$; n.s. (not significant), $p > 0.05$.

For confocal analysis of sfCac$^{GFP}$ levels and inter-sfCac$^{GFP}$ nearest neighbor distance, statistics were performed with SPSS (29.0.1.0; IBM). Normality was tested with the Shapiro-Wilk test and inspecting Q-Q plots. The Investigators were not blinded to allocation during experiments and outcome assessment. No statistical method was used to predetermine sample size. The experiments were not randomized.

## Reporting summary

Further information on research design is available in the Nature Portfolio Reporting Summary linked to this article.

## Data availability

A Reporting Summary for this article is available. The main data supporting the findings of this study are available within the article and its Supplementary Information/ Source Data file. The data/values underlying Figs. 1–8 and Supplementary Figs. S1–7 are provided as a Source Data Excel file. Specific data *p*-values are also included within the corresponding figure legends. The mass spectrometry proteomics data have been deposited to the ProteomeXchange Consortium via the PRIDE partner repository[85] with the dataset identifier PXD058345. A script for active zone segmentation analysis is available on Github [https://github.com/ngimber/BruchpilotSegmentation]. Additional details on datasets and protocols that support the findings of this study will be made available by the corresponding author. Source data are provided with this paper.

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

## Acknowledgements

We acknowledge the assistance of the core facility BioSupramol supported by the DFG and the research center SupraFAB. We thank Ryota Fukaya and Hanns-Eugen Stöffler (Mönsheim; Germany) for comments; Berit Söhl-Kielczynski and Christian Rosenmund for providing us access to the HPF machine and for support in performing the HPF experiments; Heike Stephanowitz and Max Ruwolt for excellent technical assistance in performing mass-spectrometry experiments. This work was supported by the European Commission (ERC Advanced Grant "SynProtect" to S.J.S), and by grants from the Deutsche Forschungsgemeinschaft (DFG) to S.J.S: CRC1315/A08 (project ID 327654276), FOR5289 (project ID 453877723), FOR5228 (project ID 447288260); FOR2705 (project ID 365082554); Excellence Cluster Neurocure (EXC-2049-390688087) and NeuroNex2 (project ID 436260754). This work was supported by grants from the DFG to A.M.W. (Transregio SFB 186, project ID 278001972) and the Novo Nordisk Foundation (Young Investigator Award project ID NNF19OC0056047 to A.M.W.).

## Author contributions

Conceptualization: J.L., T.M.R., S.J.S. Methodology: J.L., T.M.R., S.J.S. Investigation: J.L., T.M.R., S.L., T.G., L.G., O.T., M.M., M.G., S.P., N.R., T.G., A.N., J.S., A.S. Formal analysis: J.L., T.M.R., S.L., T.G., L.G., O.T., M.M., M.G., S.P., N.R., T.G., A.N., N.G., K.L., M.H., B.B., M.H., F.L. Visualization: J.L., T.M.R., M.M., N.R., S.J.S. Resources: D.B., T.M., J.S., A.M.W., M.H., K.L., F.L., M.C.W., S.J.S. Funding acquisition: S.J.S. Project administration: J.L., T.M.R., S.J.S. Supervision: S.J.S. Writing—original draft: S.J.S. Writing—review and editing: S.J.S., A.M.W.

## Funding

## Competing interests

The authors declare no competing interests.
