## [Transparent Peer Review file · Nature Communications]

Blobby is a synaptic active zone assembly protein required for memory in *Drosophila*

Corresponding Author: Professor Stephan Sigrist

Version 0:

Reviewer comments:

Reviewer #1

(Remarks to the Author)

The manuscript by Lutzkendorf et. al. describes the identification and characterization of a new synaptic active zone (AZ) protein at *Drosophila* neuromuscular synapses. Using immunoprecipitation with anti-BRP antibodies to pull down a known AZ protein, the authors identify a number of hits, including CG42795, which they term Blobby. The authors generate antisera to the protein to show it resides at presynaptic AZs, confirmed by genomic tagging in motoneurons. The authors then generate CRISPR null mutants of the locus and characterize the resulting synaptic defects. The mutants show reduced NMJ growth and fewer AZs. In addition, NMJs in mutant animals often show 2 to 6 abnormal accumulations of BRP (Figure 3J) that lack postsynaptic glutamate receptor staining, suggesting ectopic BRP clustering can happen at a low frequency in the absence of Blobby. Electrophysiology reveals reduced evoked responses and spontaneous mini amplitudes, as well as increases in paired-pulse facilitation. Using EM, the authors find that the structure of the AZ T-bar is disrupted, with a loss of the elongated bar platform typical of fly AZs. However, SV number and docking at AZs appears unaffected. The authors then use intravital imaging of tagged presynaptic calcium channels (Cac-mEOS) to document a change in channel clustering and density within the AZ, as well as a decrease in channel mobility, which they suggest may underlie the electrophysiological defects in the mutant. Overall, the paper describes a new AZ protein that will be of broad interest to the field. However, the authors should expand the discussion of their data to include other possibilities for what Blobby may be doing. They focus on a possible role in liquid-liquid phase separation of BRP as a mechanism, but this is a relatively modest phenotype, with only a few blobs per NMJ that seem unlikely to account for most of the phenotypes. A few suggestions to improve the discussion and some additional comments on results they did not highlight but should, are noted below.

Comments:

1. The authors completely fail to discuss any behavioral phenotypes of the Blobby mutant. Is the mutant lethal, or do normal adults emerge. If so, do the adults show any motor defects as seen in BRP null mutants? If not, what do the authors conclude from the lack (or presence if they are there) of behavioral phenotypes?
2. One phenotype that appears throughout the images in Blobby mutants is a reduction in GluRIID PSD staining compared to controls (Figure 3). The authors need to quantify GluRIID staining across the AZ population to document this decrease per AZ. Coupled with the almost 50% decrease in mini amplitude that the authors do not mention in the text (again consistent with a large decrease in postsynaptic GluRs), and the ~30% decrease in AZ number, one could conclude that the defect in evoked response can almost entirely be explained by these two phenotypes. The authors ignore these in their discussion and focus on BRP liquid phase separation and calcium channel disruption. The authors should comment on these changes in both the results and discussion and indicate why they favor a more exotic explanation for the physiology defects rather than the two obvious ones mentioned above.
3. Have the authors considered the possibility, similar to other known TBC/GAP proteins of this family, that Blobby may act as a GAP for Rab3 or other synaptic Rab proteins that contribute to the synaptic defects?
4. In relation to the above, Rab3 mutants specifically show a defect in accumulation of late scaffold proteins versus early scaffolds, as shown by the authors. In that regard, do the BRP and RBP-containing Blobs that appear in synaptic boutons also contain early scaffold proteins like Liprin, Syd or Unc13B? Similarly, do the authors see any early AZ scaffold sites apposed to GluRs that lack BRP and RBP in Blobby mutants as found in Rab3 mutants.

5. Do the authors know when Blobby appears at AZs during developmental maturation – does it arrive with BRP and other late scaffolds, or does it show up before BRP with the early scaffold proteins. Knowing this timing might provide insights into what the protein is doing to organize AZ scaffolds overall.

6. The authors should include a supplemental table with all their BRP immunoprecipitated hits that they show a graph of in Figure 1A. There are many unlabeled spots throughout what should be the BRP enriched protein pull-downs, so it is difficult to know how robust this approach was without knowing the identity of the other proteins enriched in the anti-BRP immunoprecipitation. Clearly, Blobby is an AZ enriched protein at the fly NMJ based on staining, but it would be good to know what the other hits are.

7. The authors overuse the word “obviously” – please remove most of these as they are not useful.

Reviewer #2

(Remarks to the Author)

In this study the authors describe the expression of a novel protein Blobby using elegant state-of-art methods. The major findings are that:

1. Blobby localizes in close proximity to Bruchpilot (Brp) and RIM-binding protein (RIM-BP) at the active zone (AZ) scaffold of the *Drosophila melanogaster* motoneurons.
2. Brp knock-out disturbs assembly of blobby in AZ.
3. Blobbynull and blobby-STOP-ALFA show smaller NMJs and thus fewer AZs but with the same number of AZs per NMJ.
4. In both KO genotypes, Brp is no further aligned with post-synaptic ionotropic glutamate receptors and presents conglomerates of Brp, assigned as blobbs
5. Deletion of blobby resulted in smaller evoked and miniature EJCs, with smaller quantal content and increased PPRs.
6. STEM microscopy revealed dimorphic AZs with larger perimeters.

The technological and scientific merits of the study seem excellent. The paper is very well written and illustrated and the conclusions and data interpretation are valid and reliable, however, multiple points in the data presentation and methodology need clarification as described below.

Comments:

- 1) Can you please quantify the appearance of the `blobbs`. In how many NMJs per larvae are these blobs visible? Is there a difference in blobbynull or the blobby-STOP-ALFA genotypes.
- 2) Can a cross-reactivity of the rabbit polyclonal Blobbyex8b antibody and the BRPNC82 monoclonal antibody be completely excluded?
- 3) Did the authors undergo a mistake in the legends of Fig. 4J-P? The sizes of the eEJC amplitudes etc. do not fit with the current sizes in the example traces. In P are the mean \pm SEM bars missing.
- 4) Why are the mEJC amplitudes twice as large in the wt118 and blobbynull mutants compared to the ok6> blobby-ALFA and blobby-STOP-ALFA larvae?
- 5) The authors suggest that blobby plays a role in increasing release probability indicative of the increased PPR and smaller initial amplitudes. A deeper characterization of the electrophysiology would be necessary to better understand the mechanisms of Blobby:
 - a) Please quantify the kinetics of the miniature and evoked EJCs, as the release probability can have an impact on the rise time of the EJCs as occurred in your previous investigation on Brp (Kittel et al., Science 2006). The missing overlap of Brp and GluR2D might influence the decay times.
 - b) Reduced EJC amplitudes may evolve from a reduced release probability or a reduced number of release sites. Please perform a multiple probability fluctuation analysis after applying different calcium concentrations during eEJCs to analyze whether pre- or N are reduced, as performed previously by your group (Reedy-Alla et al., Neuron 2017). If pre- is reduced, please conduct experiments with calcium chelators to work out, whether a potentially reduced pre- results out of a different coupling distance between SVs and voltage-gated calcium channels.
- 6) How do the AZ positions are altered in blobbynull mutants quantified in STED microscopy. What does a nearest-neighbor of AZ analysis reveal? #
- 7) Please give the unit of perimeter of acquired gSTED images in Fig. 5.
- 8) What are the structural correlates in TEM after high-pressure freezing of `blobbs` seen in fluorescence imaging?
- 9) In Fig. 5G two giant-clear core SVs are visible in the vesicular pool in blobbynull mutants. Is this quantitatively verifiable, whether a lack of blobby leads to an accumulation of giant-clear core SVs and may this have an implication for release and ultimately for short-term plasticity, as suggested in Maus et al. 2020 (<https://doi.org/10.1016/j.celrep.2020.02.083>)?
- 10) EM-analyses: Quantification of T-bar numbers in Figure 5F and the synaptic vesicle densities in Figure S2C and Fig 5H needs to be described in more detail. On how many serial sections were the T-bars identified? How was the analysis of the synaptic vesicle pools performed? Was HPF tissue used and what type of EM imaging was used? No differences were found within the SV pools at 0-5 nm and 0-10nm from the AZ. In the representative images of Fig. 5G, however, the vesicle pools in the Blobby-KO seem to be floating away of the tightly docked vesicles. Could this be further quantitated, even if an accurate quantitation of vesicle pools would require EM-tomography. In the methods part EM sample processing including fixation with glutaraldehyde and HPF/ freeze substitution etc. should be described as well as the 2D EM imaging analyses.
- 11) Live Cac-mEOS4b imaging of type 1b NMJs was performed from segments A2-A4 on Muscle 4 or 6/7). Did you detect differences in distribution or density in muscle 4 vs 6/7? In the methods, please provide references for the dSTORM and

QPAINT imaging studies used for selecting the parameter set for Cac localizations.

12) Statistics: All experiments in Fig. 4 are statistically compared with an unpaired t-test. Please provide evidence, whether all data samples are normally distributed throughout the manuscript, if they were compared with a t-test or an ANOVA test. Please indicate, why you have chosen to statistically test the data in Fig. 5F and H using a Kolmogorov Smirnov test rather than a Mann Whitney U test?

Reviewer #3

(Remarks to the Author)

Reviewer #4

(Remarks to the Author)

General comments:

In this manuscript, the authors identify Blobby as a novel regulator of the active zone architecture. The authors find Blobby in an immunoprecipitation screen for BRP interactors. Using *Drosophila* neuromuscular junction as a model synapse, they show that loss of Blobby causes a reduction in the number of active zones and an ectopic aggregation of scaffold proteins. This correlates with defects in synaptic currents and synaptic vesicle release probability. These findings are relevant for the field, but a significant number of clarifications are needed in order to strengthen the conclusions. Furthermore, the paper is very difficult to follow and not well-written.

Major comments:

0. Please work on the text to make this more readable and accessible to the reader.

1. To confirm the interaction of BRP and Blobby, the authors should repeat the co-IP with anti-BRP antibody and assess if Blobby is present in the eluate by western blot, and if possible, perform the co-IP with one of their novel anti-Blobby antibodies and show the blot for BRP.

2. The authors can include in a supplementary figure the elegant experiment in which they express the KD recombinase in the muscle to confirm that Blobby signal is presynaptic (currently stated as not shown).

3. For the quantification of the colocalization of Blobby and RIM-BP, I assume the authors did the statistical analysis on the n number of boutons and not on the N number of animals, as the graphs represent the co-localization coefficients for all the boutons. However, considering the big spread of the co-localization coefficients in the BRP null mutants, the analysis should also be performed on the means of each animal and these should also be depicted in the graph (this will clarify whether the effect is driven by the reduced co-localization values of all the boutons from a single animal). The same applies for the rest of quantifications performed in the manuscript.

4. The "blobby" phenotype, although interesting, seems to be restricted to very few boutons. Some sort of quantification of the number and size of the blobs observed in different animals would be useful.

5. The whole interpretation of the electrophysiological defects based on the single molecule imaging of Cacophony data is confusing: although significant (see comment 3), the differences observed in Blobby null mutants seem too small to be biologically relevant. I suggest this hypothesis (Blobby controlling the location of calcium channels at the active zone) is either further explored with additional experiments (preferred) or this section is removed from the manuscript.

6. Since most of the discussion speculates on a potential role of Blobby in liquid-liquid phase separation (a hot and fashionable topic), this should be experimentally tested or other alternative explanations (eg direct interaction with other scaffold proteins) should be also discussed.

Minor comments:

1. page and line numbers are helpful for reviewers.

2. Merge the last two paragraphs in page 3 and delete the word however.

3. The sentence "the Blobby signal at presynaptic AZs is obviously of motoneuron means presynaptic origin" seems incorrect.

4. Error in the legend of Fig. 2E,F: RIM-BP instead of BRP.

5. Delete the word however at the end of page 4 after notably.

6. on page 6, paragraph 3, delete the word however.

7. Avoid describing HPF-EM technicalities and advantages in the main text (page 6).

8. Incorrect figures referred to on numerous occasions in the text (no figure number in the beginning of page 6, figure 4 instead of figure 5 later in page 6...). These are avoidable issues...

Reviewer #5

(Remarks to the Author)

In this manuscript the authors identify and characterise a new protein that interacts with AZ proteins to facilitate SV trafficking. To identify this perviously unannotated protein, the authors performed a Co-IP LC-MS experiment. The details surrounding these methods are very brief in the manuscript and require further explanation. Below are some specific comments:

1. No details were provided in the methods section regarding the BRP IP, only references to published literature. It is okay to lean on published data to support the MM section, but a description of the work at a basic level is still necessary. At present

the method cannot be evaluated or reproduced from this manuscript.

2. For the LC-MS analysis, a gel was run and in-gel tryptic digest performed. There is no information on the bands that were excised. Was it a brief gel run only for clean up? Or were proteins separated and digested in independent slices? There is insufficient information provided to understand or repeat the experiment.

3. There is insufficient information provided for the MS acquisition and analysis. What MS instrument was used to collect the data? What was the mass tolerance used during database searching? What modifications were set in the database searching?

4. In the volcano plot analysis and S_{0} value and FDR cut off are described, but these criteria are not applied in the data visualisation. Here only a non-adjusted P value is used, displaying no FDR correction. Additionally, the volcano plot appears truncated on the x-axis. It is better to show the complete data profile in the plot.

5. In the provided raw data table, I was unable to locate CG42795 in any searchable column (FASTA head or gene name). What is the associated protein identifier or gene name for this protein? The supporting data should be easily interpretable for the reader to look at. Additionally, I could not find the cited Unc-13A in the MS results, but rather Unc-13 as a master protein. There did not appear to be any evidence supporting the A isoform as the predominant protein detected.

Version 1:

Reviewer comments:

Reviewer #1

(Remarks to the Author)

The authors have addressed my prior concerns.

Reviewer #2

(Remarks to the Author)

The authors have provided satisfactory responses to all points raised in the previous review. As proposed the data in reviewer Figure 2 with the quantitation of blobs in wt and blobbynull NMJs would be useful to be integrated into the manuscript.

Reviewer #3

(Remarks to the Author)

Reviewer #4

(Remarks to the Author)

The authors have addressed all my concerns and I have no further questions. this is a nice contribution to NCOMMS

Reviewer #5

(Remarks to the Author)

The authors have addressed all of my concerns raised in the manuscript review. They have provided the requested information to a satisfactory degree.

Specific answers to the points raised by the reviewers:

Reviewer 1:

Overall, the paper describes a new AZ protein that will be of broad interest to the field. However, the authors should expand the discussion of their data to include other possibilities for what Blobby may be doing. They focus on a possible role in liquid-liquid phase separation of BRP as a mechanism, but this is a relatively modest phenotype, with only a few blobs per NMJ that seem unlikely to account for most of the phenotypes. A few suggestions to improve the discussion and some additional comments on results they did not highlight but should, are noted below.

We are happy to hear that the reviewer considers our work to be of broad interest to the field. We also agree with the critique and changed the manuscript accordingly as detailed below.

Point-to-point response:

1. *The authors completely fail to discuss any behavioral phenotypes of the Blobby mutant. Is the mutant lethal, or do normal adults emerge. I so, do the adults show any motor defects as seen in BRP null mutants? If not, what do the authors conclude from the lack (or presence if they are there) of behavioral phenotypes?*

We now performed behavioral analysis, and indeed can demonstrate significant behavioral relevance of Blobby. Analyzing the relevance of Blobby for the vitality of adult flies, we first determined eclosion rates in several independent experiments, which turned out to be significantly and robustly reduced (Fig. 8A). Measuring locomotion using a negative geotaxis assay, a severe, significant reduction of adult *blobby*^{Null} mutant animals relative to controls was observed. Also these data have now been integrated into the manuscript (Fig. 8B).

We then studied a putative role of Blobby in post-developmental mushroom body mediated learning and memory. To this end, we used a somatic CRISPR approach knocking down Blobby expression specifically in the intrinsic neurons of the mushroom body ("Kenyon cells"). This knockdown further was restricted to the adult post-developmental post-hatching stage using the *gal80ts* strategy (shifting animals to 31°C to leave the *gal80* block). Notably, we observed a near absence of consolidated aversive olfactory memory measured at 1 hr or 3 hrs after training (Fig. 8E, G; 18°C controls show that the phenotype critically depends on the knockdown Fig. 8F, H). At the same time, initial learning (STM) was not affected (Fig. 8D). This result aligns interestingly with our recent work, which shows that BRP/AZ remodeling within Kenyon cells is specifically required for mid-term memory formation ^{1,2}.

2. One phenotype that appear throughout the images in *Blobby* mutants is a reduction in *GluRIID* PSD staining compared to controls (Figure 3). The authors need to **quantify *GluRIID* staining across the AZ population to document this decrease per AZ.**

We did now quantify *GluRIID* staining levels (Reviewer Fig. 1).

As shown in Reviewer Figure 1, we observed a tendency towards a difference; however, this difference did not reach statistical significance. Therefore, we did not emphasize this trend in the revised manuscript. Should the reviewer consider it important, we are open to incorporating it.

- 2.1 Coupled with the almost 50% decrease in mini amplitude that the authors do not mention in the text (again consistent with a large decrease in postsynaptic *GluRs*), and the ~30% decrease in AZ number, one could conclude that the defect in evoked response can almost entirely be explained by these two phenotypes.

We appreciate this very valid point of the reviewer. We now clearly state that the reduced number of AZs per NMJ terminal directly contributes to the reduced evoked release observed by TEVC measurements in both investigated genotypes (Fig. 6).

We do now write in lines 247 to 250:

“Instead, our data suggest that a reduction in the total number of AZs per NMJ terminal (Fig. 3C), along with a decreased SV release probability, indicated by the robustly increased paired-pulse responses (Fig. 6D, E; 6L, M; Fig. S6II F, G), accounts for the observed release deficits.” And in lines 265 to 267: “Taken together, mean-variance analysis confirms that both, a reduced number of SV release sites but also deficits in SV release probability underlie the release deficits of *blobby* mutants at physiological Ca^{2+} concentrations.”

The reviewer rightfully mentions the reduced mini amplitude of the $blobby^{Null}$ animals. We now mention this in lines 213-214:

“Additionally, mEJC amplitudes were significantly reduced relative to controls in *blobby*^{Null} animals (Fig. 6G).”

This reduction of postsynaptic sensitivity in all likelihood contributes to the eEJC reduction of *blobby*^{Null} animals as well. We would like to emphasize, however, that quantal contents in both Blobby-deficient genotypes displayed a strong and significant reduction (Fig. 6C, K). Thus, the number of SVs released per action potential (calculated by dividing eEJC values by the respective miniature excitatory junctional amplitudes) is severely reduced irrespective of a lack of postsynaptic glutamate sensitivity.

2.2 The authors ignore these in their discussion and focus on BRP liquid phase separation and calcium channel disruption. The authors should comment on these changes in both the results and discussion and indicate why they favor a more exotic explanation for the physiology defects rather than the two obvious ones mentioned above.

We understand this comment of the reviewer, which is very well taken. We completely agree that the reduction of evoked release has in all likelihood a dominant contribution from the reduced number of release sites, and are sorry not to have made this clear in the first version (see above). To validate this interpretation, we now performed a mean variance analysis (Fig. 7), which found that indeed N, the number of functional release sites, is reduced in *blobby*^{Null} (Fig. 7F). In the same analysis, a change of SV release probability was observed as well, in an interesting relationship to the increased paired pulse ratios we robustly measured for *blobby*^{Null} (Fig. 6D,E) and in the stop-cassette animals (Fig. 6L,M). As indicated above (comment 2.1), we now mention that the reduced number of AZs per NMJ terminal directly contributes to the reduced evoked release observed by TEVC measurements in both investigated genotypes.

We still think that the nanoscale patterning defect of Ca²⁺ channels we described using single molecule imaging (sptPALM) in the previous first version of the manuscript might contribute to the SV release probability deficits, which we now validated by mean variance analysis. However, to really establish unequivocal causal connections here will need quite extensive experimentation and modeling. Thus, as the reviewers remain rightfully skeptical concerning the role of Ca²⁺ channel distribution deficits, we now decided to remove the Ca²⁺ channel single molecule data from the manuscript (also see reply to reviewer #4 below). Concerning a putative role of Blobby for AZ scaffold liquid-liquid phase separation: we did not mean to explain specific physiological deficits, but that it might be responsible for the mistargeting and defective nanopatterning of AZ scaffold material. It is also tempting to speculate that Blobby binding might be needed to render BRP/RIM-BP into a "consumable state," possibly by exposing critical

interaction surfaces needed for their binding to early scaffolding components. To avoid further confusion, however, we now restrict its discussion to a few sentences in the discussion.

3. *Have the authors considered the possibility, similar to other known TBC/GAP proteins of this family, that Blobby may act as a GAP for Rab3 or other synaptic Rab proteins that contribute to the synaptic defects?*

We indeed in the moment test the possibility that the TBC Rab-GAP domain of Blobby might control Rab3 or other AZ Rabs and thus contribute to the blobby phenotypes. However, we do feel that the current manuscript already presents a lot of data and discovery. Also given the early time point of our analysis, we would prefer to not include any additional data in this direction.

1. *In relation to the above, Rab3 mutants specifically show a defect in accumulation of late scaffold proteins versus early scaffolds, as shown by the authors. In that regard, do the BRP and RBP-containing Blobs that appear in synaptic boutons also contain early scaffold proteins like Liprin, Syd or Unc13B? Similarly, do the authors see any early AZ scaffold sites apposed to GluRs that lack BRP and RBP in Blobby mutants as found in Rab3 mutants.*

This is a very interesting point. We have now included a STED analysis of two early scaffold proteins, Syd-1 as well as Unc13B, into the revised manuscript (Fig.4 I, J). Unfortunately, Liprin- α could not be used for this analysis as it is expressed at both pre- and postsynaptic specializations, making clear assignment of the label difficult. Notably, both early scaffold proteins do not incorporate into the BRP Blobs. This aligns with our new intravital imaging analysis, which shows that Blobby follows BRP during AZ assembly (Fig. 2G, H). Therefore, Blobby may play a specific role in ensuring the proper transfer and integration of late BRP complexes into developing AZ scaffolds. We have now revised the discussion to address this point accordingly.

2. *Do the authors know **when Blobby appears at AZs during developmental maturation** – does it arrive with BRP and other late scaffolds, or does it show up before BRP with the early scaffold proteins. Knowing this timing might provide insights into what the protein is doing to organize AZ scaffolds overall.*

This was an excellent suggestion. We have now performed intravital live imaging of Blobby together with BRP. This analysis indicates that Blobby co-integrates “late” into newly forming NMJ AZs (Fig. 2G, H).

3. *The authors should include a supplemental table with all their BRP immunoprecipitated hits that they show a graph of in Figure 1A. There are many unlabeled spots throughout what should be the BRP enriched protein pull-downs, so it is difficult to know how robust this approach was without knowing the identity of the other proteins enriched in the anti-BRP immunoprecipitation.*

This was a very good suggestion. We now prepared a table (supplemental table I) which lists all the immunoprecipitated hits accordingly.

4. *The authors overuse the word “obviously”; please remove most of these as they are not useful.*

Thanks for the comment, we obviously overused obviously ☺. We now eliminated the word where not helpful.

Reviewer #2 (Remarks to the Author):

The technological and scientific merits of the study seem excellent. The paper is very well written and illustrated and the conclusions and data interpretation are valid and reliable, however, multiple points in the data presentation and methodology need clarification as described below.

We appreciate the reviewer's positive feedback on the technological and scientific merits of our study. We are pleased that the writing, illustrations, and the validity and reliability of our conclusions and data interpretation were well-received. In response to the comments regarding clarification of the data presentation and methodology, we have made significant efforts to address these concerns and have included substantial new data to enhance clarity and strengthen our findings.

Point-to-point response:

1. *Can you please quantify the appearance of the `blobbs`. In how many NMJs per larvae are these blobs visible? Is there a difference in blobby^{Null} or the blobby-STOP-ALFA genotypes.*

We apologize for not having described our analytical strategy more clearly in this instance. The term "blob" was our attempt to label the ectopic, irregular scaffold material we observed. "Blobs" refer to ectopically localized BRP-containing scaffold material, which are heterogeneous in size and vary in shape and "fuzziness". For our initial submission, we preferred to quantify ectopic AZ scaffold material based on the absence of adjacent GluRIID signal (Fig. 3E, J). However, we understand the reviewer's request to quantify how often blobs occur at individual terminals. We have now counted individual blobs using a specific size criterion. Ectopic material with a diameter greater than 0.3 μm were considered as a blob (see Reviewer Fig. 2). We specifically analyzed muscle 4 NMJs: here 4-5 NMJs from total 10 NMJs analyzed per larva contained a blob with a size greater than 0.3 μm . We are prepared to integrate these data into the manuscript if considered appropriate.

Reviewer Fig. 2 Definition and quantification of blobs.

Quantification of ectopic BRP-positive material at third-instar larval muscle 4 NMJs. Regions of interest (ROIs) for each identified BRP-positive spot were generated using ImageJ. Ectopic BRP material was quantified by calculating the ratio of BRP^{NC82} intensity to GluIID intensity within each ROI. The cutoff for ectopic material was defined as the sum of 3x the standard deviation plus the mean of these ratios. Ratios exceeding this cutoff were classified as ectopic material, and BRP-positive ectopic material larger than 0.3 μm (dotted line) was defined as a 'blob.'

A) Quantification for *blobby*^{Null} (*w*¹¹¹⁸ 0.11 \pm 0.01, n=83; *blobby*^{Null}: 0.66 \pm 0.07, n=number of AZs).

B) number of NMJs per animal containing blobs (i.e. ectopic material greater than 0.3 μm) *w*¹¹¹⁸ 0 \pm 0 % n=5; *blobby*^{Null}: 84.03 \pm 4.36 %, Mann-Whitney-U test. N=6. N=number of animals.

2. Can a cross-reactivity of the rabbit polyclonal *Blobby*^{ex8b} antibody and the BRP^{NC82} monoclonal antibody be completely excluded?

Whether any cross-reactivity to the usually co-stained BRP^{NC82} monoclonal antibody can be excluded and whether indeed the BRP AZ scaffold staining is "real" is certainly a very important question. We can indeed safely exclude unspecific cross-reactivity. First, we exclusively stained for *Blobby* (with *Blobby*^{C-Term}) not using a co-labeling for another AZ protein (Fig. 1E upper lane; co-label here against the diffusely expressed HRP) and simultaneously documented antibody specificity (Fig. 1E, lower lane). Second, we specifically detected *Blobby* AZ label (with *Blobby*^{ex8b}) via an ALFA-Tag staining after recombining out a STOP-cassette from the *blobby*-STOP-ALFA stock (Fig. 2B).

3. *Did the authors undergo a mistake in the legends of Fig. 4J-P? The sizes of the eEJC amplitudes etc. do not fit with the current sizes in the example traces. In P are the mean ± SEM bars missing.*

We are grateful for this careful review. We corrected the indicated mistakes.

4. *Why are the mEJC amplitudes twice as large in the wt1118 and blobby^{Null} mutants compared to the ok6 blobby-ALFA and blobby-STOP-ALFA larvae?*

This is a very valid point. When comparing the genotypes shown in old Fig. 4A-H with those in old Fig. 4I-P (now Fig. 6A-H, I-P), it is important to note that they involve different genetic backgrounds. In particular, both genotypes in old Fig. 4I-P (now Fig. 6 I-P) include the *ok6-gal4* driver line. This driver line, as well as others, can indeed have noticeable effects. This also could potentially explain why the mEJC frequencies tend to be generally higher in these genotypes (please compare old Fig. 4H with Fig. 4P, now Fig. 6H and P).

We would like to emphasize that in all experiments, we compared the *blobby^{Null}* mutant to the isogenic background that we used for its CRISPR-based generation.

5. *The authors suggest that blobby plays a role in increasing release probability indicative of the increased PPR and smaller initial amplitudes. A deeper characterization of the electrophysiology would be necessary to better understand the mechanisms of Blobby: a) **Please quantify the kinetics of the miniature and evoked EJCs, as the release probability can have an impact on the rise time of the EJCs as occurred in your previous investigation on Brp (Kittel et al., Science 2006). The missing overlap of Brp and GluR2D might influence the decay times.***

The reviewer raised a valid point regarding the relation of *blobby* and *brp* phenotypes and the differences in release kinetics observed here. As suggested, we have now quantified the kinetics of both miniature and evoked EJCs (Fig. S6I). In the *blobby^{Null}* mutant, we observed a slowdown in both evoked and miniature kinetics (Fig. S6A-D). However, this phenotype was not consistently observed in the *ok6*, *blobby-STOP-ALFA* (control) versus *ok6>blobby-ALFA* (mutant) experiment (Fig. S6F-I). These differences might well be explained by differences in the genetic background between the two genotypes as well. For this reason, we have chosen not to emphasize findings in regard to release kinetics in the manuscript.

We would like to emphasize, however, that the reduced eEJC amplitude and paired-pulse facilitation (at both 10 ms and 30 ms interpulse intervals) were significantly and consistently observed across all recordings, both in the *blobby^{Null}* mutant (Fig. 6B,D,E) and the *ok6*, *blobby-STOP-ALFA* mutant (Fig. 6J,L,M).

*b) Reduced EJC amplitudes may evolve from a reduced release probability or a reduced number of release sites. **Please perform a multiple probability fluctuation analysis after applying different calcium concentrations during eEJCs to analyze whether P_{rel} or N are reduced, as performed previously by your group (Reedy-Alla et al., Neuron 2017).***

We have precisely followed the reviewer's insightful suggestion and performed a mean variance analysis (new Fig. 7). Indeed, we here show that both P_{rel} and N are reduced in the absence of Blobby (Fig. 7D,F). Our conclusion is: "Taken together, mean-variance analysis confirms that both, a reduced number of SV release sites but also deficits in SV release probability underlie the release deficits of blobby mutants at physiological Ca^{2+} concentrations." (lines 265-267).

If P_{rel} is reduced, please conduct experiments with calcium chelators to work out, whether a potentially reduced P_{rel} results out of a different coupling distance between SVs and voltage-gated calcium channels.

Given that we observed a reduction in P_{rel} , and following the reviewer's suggestion, we conducted buffering experiments using an EGTA chelator (Fig. S6II). If increased coupling distances were present, we would expect a heightened sensitivity in the evoked release of the *blobby*^{Null} mutant. However, in our new recordings of *blobby*^{Null} release with and without EGTA-AM ester (following the protocols of Kittel et al., Science 2006 and Liu et al., Science 2011), we did not observe any signs of increased EGTA sensitivity. If anything, there was a slight trend towards decreased sensitivity. Since the Cac Ca^{2+} channel clusters mark the center of individual AZs, we took this opportunity to measure AZ-AZ distances. The Cac-Cac cluster distances were not increased, indicating an essentially unchanged AZ density at *blobby*^{Null} NMJ terminals (Fig. 4F), consistent with our confocal measurements of AZ density based on BRP signals (Fig. 3D).

We conclude in lines 245-250: "In summary, we found no evidence of deficits in the nano-spacing between Ca^{2+} channels and SV release sites or in the targeting of SVs to the AZ plasma membrane as contributors to the functional deficits in blobby mutants. Instead, our data suggest that a reduction in the total number of AZs per NMJ terminal (Fig. 3C), along with a decreased SV release probability, indicated by the robustly increased paired-pulse responses (Fig. 6D, E; 6L, M; Fig. S6II F, G), accounts for the observed release deficits."

6. *How do the AZ positions are altered in *blobby*^{Null} mutants quantified in STED microscopy. What does a nearest-neighbor of AZ analysis reveal?*

Thank you for the excellent suggestion, which we have followed experimentally (Fig. 4E, F). As an ideal measure, we selected the AZ-central Cacophony clusters using gSTED microscopy. We measured the Cac-Cac cluster distances. While the differences between *blobby*^{Null} and control were only small, we observed a slight reduction in the intra-AZ cluster distances in Blobby-lacking NMJ terminals.

We now write in lines 164-168: "Since the Cac Ca²⁺ channel clusters mark the center of individual AZs, we took this opportunity to measure AZ-AZ distances (see Material and Methods). The Cac-Cac cluster distances were not increased, again suggesting an essentially unchanged AZ density at blobbyNull NMJ terminals (Fig. 4F), consistent with our confocal measurements of AZ density based on BRP signals (Fig. 3D)."

7. *Please give the unit of perimeter of acquired gSTED images in Fig. 5.*

We apologize for this omission, we now added the unit (μm).

8. *What are the structural correlates in TEM after high-pressure freezing of `blobbs` seen in fluorescence imaging?*

This is certainly a fully justified, yet challenging question to address, as prominent blobs in fluorescence light microscopy are observed only every few micrometers along the NMJ terminals and TEM sampling. Our primary conclusion regarding the BRP scaffold concerns its apparent structural alteration, including the accumulation of ectopic material, which we believe can be sufficiently demonstrated using light microscopy. While a systematic electron microscopy study of this ectopic material would be valuable, such an investigation would require large-scale EM reconstruction of entire NMJ terminals. Given that we analyzed approximately 100 partly non-contiguous EM sections, each 60 nm thick - representing only about one-fifth to one-tenth of a full NMJ length - this is beyond the scope of the current manuscript. Statistical analysis of blobs based on electron microscopy will require a more comprehensive reconstruction, which we unfortunately cannot provide at this stage.

9. *In Fig. 5G two giant-clear core SVs are visible in the vesicular pool in *blobby*^{Null} mutants. Is this quantitatively verifiable, whether a lack of blobby leads to an accumulation of giant-clear core SVs and may this have an implication for release and ultimately for short-term plasticity, as suggested in Maus et al. 2020.*

We greatly appreciate the reviewer's keen observation and thoughtful suggestions. In response, we have now systematically analyzed two distinct size classes of vesicles as a

function of their vertical distance from the AZ plasma membrane (Fig. 5 F,G). Vesicles were categorized as 40 nm or smaller (to capture synaptic vesicle information) and larger, clear vesicles, and counted in bins up to a vertical distance of 400 nm. Our analysis did not reveal any differences between for vesicles smaller 40 nm. Concerning ("giant") clear vesicles, we indeed observed a tendency towards increased numbers in more distant bins densities in *blobby*^{Null} (Fig. 5G). As large giant vesicles were not increased close to the membrane, however, we now in Fig. 5C (old Fig.5G) replaced the HPF *blobby*^{Null} image with a more representative one.

10. EM-analyses: Quantification of T-bar numbers in Figure 5F and the synaptic vesicle densities in Figure S2C and Fig 5H needs to be described in more detail. On how many serial sections were the T-bars identified?

We appreciate the reviewer's comment and apologize for not providing a more detailed description of our approach earlier. This opportunity allows us to clarify our methodology.

Regarding the T-bar analysis, we would first like to emphasize that STED microscopy revealed a deficit in the distribution of the major T-bar component, BRP. The subsequent T-bar shape analysis was conducted to provide independent confirmation of this finding. It is important to note that T-bars are most readily observed using conventional embedding procedures, which involve sample dehydration.

To estimate T-bar numbers and morphologies, we quantified ideally imaged AZ cross-sections with at least 400 nm planar membrane contacts between the motoneuron and muscle plasma membrane (see Fig. 5A). We visually identified T-bars by the presence of both a pedestal (> 80 nm) and a roof structure (> 150 nm), which from our experience with contiguous sections is a reliable criterion for T-bar identification. During sampling, we ensured not to double-count T-bars in contiguous sections.

Using these criteria, we observed fewer T-bars at *blobby*^{Null} terminals (Fig. 5A-B), which aligns with the defective BRP organization observed through STED microscopy (Fig. 4A).

11. How was the analysis of the synaptic vesicle pools performed? Was HPF tissue used and what type of EM imaging was used? No differences were found within the SV pools at 0-5 nm and 0-10nm from the AZ.

Regarding the analysis of the synaptic vesicle pools, we indeed used HPF tissue, as it reliably preserves vesicle distribution (though it provides rather limited contrast for active zone specializations such as T-bars). We systematically analyzed AZ cross-sections (characterized by larger than 400 nm planar membrane contacts between the motoneuron and muscle plasma

membrane, which was clearly visible in the HPF analysis) and counted synaptic vesicles (smaller than 40 nm) near the motoneuron plasma membranes. The results of this analysis are seen in Fig. 5D-E. As reported in our previous submission, the numbers and densities of SVs (< 40 nm) were not significantly different between *blobby*^{Null} mutants and controls (Fig. 5D-F). This suggests that deficits in SV docking due to defective SV transfer to the AZ membrane are not primarily responsible for the Blobby deficit.

Following the reviewer's suggestion, we also analyzed the distribution of larger (giant) clear vesicles (> 40 nm) as a function of their distance from the AZ central membrane (across a membrane segment of 400 nm in the center of the active zone). In *blobby*^{Null}, larger clear vesicles tended to be increased at farther distances from the AZ membrane (Fig. 5G).

12. In the representative images of Fig. 5G, however, the vesicle pools in the Blobby-KO seem to be floating away of the tightly docked vesicles. Could this be further quantitated, even if an accurate quantitation of vesicle pools would require EM-tomography.

We sincerely appreciate the reviewer's observation regarding the vesicle pools in the Blobby-KO. While the initial images may have given the impression that vesicles were "floating away" from tightly docked vesicles, this was not representative of our overall dataset. To address this issue, we have replaced the image (Fig. 5C). In response to the reviewer's suggestion, we also quantified vesicle distributions from multiple sections (Fig. 5F) and found no consistent displacement of vesicle pools in the Blobby-KO that in our eyes would warrant further concern. While EM-tomography of *blobby*^{Null} AZs would in all likelihood provide additional insights, we believe our current analysis using multiple sections sufficiently captures the spatial organization of vesicle pools.

13. Live Cac-mEOS4b imaging of type 1b NMJs was performed from segments A2-A4 on Muscle 4 or 6/7). Did you detect differences in distribution or density in muscle 4 vs 6/7? In the methods, please provide references for the dSTORM and QPAINT imaging studies used for selecting the parameter set for Cac localizations.

Thank you for the justified comment. Imaging was primarily conducted on muscle 4. Importantly, even when focusing exclusively on muscle 4 data and excluding the limited muscle 6/7 data, the AZ density of Ca²⁺ channels in *blooby*^{Null} remained significantly reduced. We now provide these muscle 4-only data as Reviewer Fig. 3. Additionally, based on feedback from Reviewer #4, we have decided to remove the sptPALM data from the revised version of the manuscript. However, if this or other reviewers feel differently, we are happy to reintroduce these data as a supplementary figure and reference them in the discussion as a potential mechanism contributing to the observed release deficits.

Reviewer Fig. 3: sptPALM analysis for Cac at NMJ AZs exclusively from muscle 4.

Live sptPALM imaging of *cac*^{mEOS4b} (control) and *cac*^{mEOS4b}; *blooby*^{Null} mutants.

A) Representative boutons displaying the live cumulative distribution of Cac localization as a tessellation map of an individual synapse. Scale Bar 50nm. Tessellation analysis of B) live diffusion coefficient C) AZ diameter D) Cac localization density in AZs E) Radii of confinement F) Nanocluster (NC) diameter and G) Cac localization density in NCs, derived from the mean square displacement of single live imaged Cac molecules in *Cac*^{mEOS4b} (grey) and *Cac*^{mEOS4b}; *blooby*^{Null} (blue). PALM Data distribution was statistically tested with Kolmogorov-Smirnov test. Statistical significance is denoted as asterisks: *p < 0.05; **p < 0.01; ***p < 0.001; ****p < 0.0001.

14. *Statistics: All experiments in Fig. 4 are statistically compared with an unpaired t-test. Please provide evidence, whether all data samples are normally distributed throughout the manuscript, if they were compared with a t-test or an ANOVA test. Please indicate, why you have chosen to statistically test the data in Fig. 5F and H using a Kolmogorov Smirnov test rather than a Mann Whitney U test?*

We are happy to describe our statistical procedures. For comparisons between two normally distributed groups, we used an unpaired t-test. The Kolmogorov-Smirnov test was applied to assess differences in the shape of distributions that were not normally distributed, such as the

cumulative distribution of SVs to the plasma membrane, histograms, NMJ quantifications, and electrophysiological measurements. Based on our understanding, the Mann-Whitney U test is appropriate for comparing differences based on average ranks, but this was not relevant to our data. For comparisons involving more than two groups, we used the ordinary one-way ANOVA.

Reviewer #4 (Remarks to the Author):

General comments:

*In this manuscript, the authors identify Blobby as a novel regulator of the active zone architecture. The authors find Blobby in an immunoprecipitation screen for BRP interactors. Using *Drosophila neuromuscular junction* as a model synapse, they show that loss of Blobby*

causes a reduction in the number of active zones and an ectopic aggregation of scaffold proteins. This correlates with defects in synaptic currents and synaptic vesicle release probability. These findings are relevant for the field, but a significant number of clarifications are needed in order to strengthen the conclusions. Furthermore, the paper is very difficult to follow and not well-written.

Please work on the text to make this more readable and accessible to the reader.

We are pleased that the reviewer finds our findings relevant to the field and we sincerely appreciate the careful and thorough review of our manuscript. In response to the feedback, we have made substantial efforts in the revised version to improve the clarity and readability of the text, making it more accessible to the reader while addressing the concerns raised.

Point-to point response:

- 1. To confirm the interaction of BRP and Blobby, the authors should repeat the co-IP with anti-BRP antibody and assess if Blobby is present in the eluate by western blot, and if possible, perform the co-IP with one of their novel anti-Blobby antibodies and show the blot for BRP.*

We followed this suggestion and were able to identify Blobby in the BRP-IPs now using immuno-probing with our novel antibody Blobby^{C-term}. These data are now included in the main manuscript (Fig. 1C). Unfortunately, our Blobby antibodies did not allow for an efficient immunoprecipitation of Blobby.

- 2. The authors can include in a supplementary figure the elegant experiment in which they express the KD recombinase in the muscle to confirm that Blobby signal is presynaptic (currently stated as not shown).*

This was an excellent suggestion. We have now included these data in the supplementary material (Fig. S2). As shown, KD recombinase expression in the muscle did not produce any

Blobby staining, while motoneuron expression of KD recombinase did. Thus, the Blobby AZ signal is indeed due to presynaptic motoneuron expression.

3. For the quantification of the colocalization of Blobby and RIM-BP, I assume the authors did the statistical analysis on the n number of boutons and not on the N number of animals, as the graphs represent the co-localization coefficients for all the boutons. However, considering the big spread of the co-localization coefficients in the BRP null mutants, **the analysis should also be performed on the means of each animal and these should also be depicted in the graph (this will clarify whether the effect is driven by the reduced co-localization values of all the boutons from a single animal)**. The same applies for the rest of quantifications performed in the manuscript.

We have now conducted the statistical analysis based on the mean values per animal, as shown in the following graphs (Reviewer Fig. 4). The significance and effect size are consistent with the results obtained when the analysis was based on the number of boutons (Fig. 2E, F in the main manuscript).

Reviewer Fig. 4 The degree of colocalization between Blobby and RIM-BP is significantly reduced at brp^{Null} mutant AZs.

A, B) Quantification of overlap between BRP and Blobby^{ex8b} signals in controls and brp^{Null} mutants, based on the mean values per animal. A) Plot of Pearson's coefficients: w^{1118} 100 ± 7.28, N=4; brp^{Null} 24.51 ± 8.57, N=4); B) Plot of Manders coefficients: w^{1118} 100 ± 7.49, n=4; brp^{Null} 67.43 ± 7.50, N=4). Graphs show mean ± SEM. ***p<0.001; *p<0.01 (unpaired test was applied). N: number of animals.

The study was not designed to use animals as a standard unit. However, we understand the reviewers point. We thus re-examined our key observation using individual animals as statistical elements for electrophysiological data. Overall, the observations remain robust, despite the change in statistical unit from the individual data point to the individual animal (Reviewer Figs 5,6,7). To quantify AZ numbers in Reviewer Fig. 5A, we included a higher number of animals.

Reviewer Fig 5 AZ numbers are reduced in *bloby* mutant NMJs.

Quantification of AZ numbers of indicated genotypes. A) Number of AZs identified as discrete BRP positive clusters per NMJ, values normalized to wild type: *w¹¹¹⁸* 100 ± 7.95 n=10; *bloby^{Null}*: 65.83 ± 8.10 , n=10; *ok6>bloby-ALFA*: 100 ± 8.69 , n=5; *ok6, bloby-STOP-ALFA* 71.04 ± 5.68 , n=5, unpaired student t-test. n=number of animals.

Reviewer Fig. 6

Reviewer Fig 6 Two-electrode voltage clamp analysis of *blobby*^{NuII} mutant per animal.

(A-H) Two-electrode voltage clamp electrophysiological recordings comparing third instar larvae of *blobby*^{NuII} animals to controls. A) eEJC amplitudes (*w*¹¹¹⁸ -86.91±12.28nA, N=6; *blobby*^{NuII} -43.03±5.523nA, N=6). B) eEJC rise time (*w*¹¹¹⁸ 1.199± 0.1712ms, N=6; *blobby*^{NuII} -43.03±5.523ms, N=6). C) eEJC decay (*w*¹¹¹⁸ 4.237± 0.3049ms, N=6; *blobby*^{NuII} 5.068± 0.3990ms, N=6). D) eEJC charge (*w*¹¹¹⁸ -670.2± 138.0pC, N=6; *blobby*^{NuII} -392.1± 60.21pC, N=6). E) Paired-pulse ratio at 10 ms interpulse interval. (*w*¹¹¹⁸ 0.7450± 0.06613, N=6; *blobby*^{NuII} 1.158± 0.2130, N=6). F) Paired-pulse ratio at 30 ms interpulse interval. (*w*¹¹¹⁸ 0.9939± 0.01679, N=6; *blobby*^{NuII} 1.360± 0.05964, N=6). G) mEJC frequencies (*w*¹¹¹⁸ 1.789± 0.2358, N=6; *blobby*^{NuII} 2.183± 0.2055, N=6; *ok6>blobby-ALFA* 4.778 ± 0.4659, N=4; *ok6,blobby-STOP-ALFA* 3.689 ± 0.5921, N=5). H) mEJC amplitudes (*w*¹¹¹⁸ -0.8143± 0.009287nA, N=6; *blobby*^{NuII} -0.6090± 0.06007nA, N=6). I) Quantal contents (*w*¹¹¹⁸ 115.7± 13.83, N=6; *blobby*^{NuII} 69.88± 4.591, N=6). Graphs show mean ± SEM. An unpaired t-test was applied, *p < 0.05; ***p<0.001; ****p < 0.0001 ns = not significant. N represents the number of animals.

Reviewer Fig. 7

Reviewer Fig 7 Two-electrode voltage clamp analysis of blobby mutant NMJs.

A-I) Two-electrode voltage clamp electrophysiological recordings comparing third instar larvae NMJs of *ok6, blobby-STOP-ALFA* to *ok6>blobby-ALFA*. A) eEJC amplitudes (*ok6>blobby-ALFA* -93.42±7.804nA, N=5; *ok6, blobby-STOP-ALFA* -50.78± 1.828 nA, N=5). B) eEJC rise time (*ok6>blobby-ALFA* 1.124 ± 0.06346 nA, N=5; *ok6, blobby-STOP-ALFA* 1.208 ± 0.08136 nA, N=5). C) eEJC decay (*ok6>blobby-ALFA* 5.483 ± 0.3136 nA, N=5; *ok6, blobby-STOP-ALFA* 5.000 ± 0.3785 nA, N=5). D) Paired-pulse ratio at 10 ms interpulse interval. (*ok6>blobby-ALFA* 0.5727± 0.03053, N=5; *ok6, blobby-STOP-ALFA* 0.8281 ± 0.06908, N=5). E) Paired-pulse ratio at 30 ms interpulse interval. (*ok6>blobby-ALFA* 0.9174 ± 0.006887, N=5; *ok6, blobby-STOP-ALFA* 1.067 ± 0.03878, N=5). F) eEJC charge (*ok6>blobby-ALFA* -840.0 ± 95.31pC, N=5; *ok6, blobby-STOP-ALFA* -413.8 ± 45.86pC, N=5). G) mEJC frequencies (*ok6>blobby-ALFA* 4.778 ± 0.4659, N=4; *ok6, blobby-STOP-ALFA* 3.689 ± 0.5921, N=5). H) mEJC amplitudes (*ok6>blobby-ALFA* -0.4292 ± 0.03847 nA, N=4; *ok6, blobby-STOP-ALFA* -0.4649 ± 0.02629, N=5). I) Quantal contents (*ok6>blobby-ALFA* 221.9 ± 24.42, N=4; *ok6, blobby-STOP-ALFA* 109.1 ± 10.61, N=5). Graphs show mean ± SEM. An unpaired t-test was applied, *p < 0.05; ***p<0.001; ****p < 0.0001 ns = not significant. N represents the number of animals.

4. *The $C;blobby^D$ phenotype, although interesting, seems to be restricted to very few boutons. Some sort of quantification of the number and size of the blobs observed in different animals would be useful.*

The feedback is fully justified, and we acknowledge that other reviewers have raised similar points. We appreciate to explain our analytical strategy. The term "blob" was used to describe the ectopic, irregular scaffold material we observed, which is BRP-containing, heterogeneous in size, and variable in shape and "fuzziness." Initially, we chose to quantify ectopic AZ scaffold material based on the absence of adjacent GluRIID signal (Fig. 3E, J). However, we understand the reviewer's request for a more detailed quantification of blobs at individual terminals. In response, we have now counted individual blobs based on a specific size criterion, defining ectopic material larger than 0.3 μ m in diameter as a "blob" (see Reviewer Fig. 2). Specifically, in muscle 4 NMJs, we found that approximately 4-5 NMJs out of 10 analyzed per larva contained a blob of this size. We would be happy to incorporate these data into the manuscript if deemed appropriate.

5. *The whole interpretation of the electrophysiological defects based on the single molecule imaging of Cacophony data is confusing: although significant (see comment 3), the differences observed in Blobby null mutants seem too small to be biologically relevant. I suggest this hypothesis (Blobby controlling the location of calcium channels at the active*

zone) is either further explored with additional experiments (preferred) or this section is removed from the manuscript.

We greatly appreciate the reviewer's insightful comments and would like to clarify our position and the steps we have taken in the revised manuscript. First, we fully agree that the current data do not allow us to establish a clear causal relationship between the nano-patterning distribution deficits of voltage-gated Ca^{2+} channels, as seen in the single-molecule imaging, and the release deficits observed in the *blobby* mutant. Addressing this thoroughly would require substantial additional experimentation and modeling, which is beyond the scope of this revision, particularly given that this is the first description of a novel AZ assembly protein.

Based on these considerations, we have followed the reviewer's suggestion and removed this data from the revised manuscript. However, we are open to reconsidering this decision based on the feedback of other reviewers and the editor.

We do believe that the subtle Ca^{2+} channel density deficits observed could still contribute to the *Blobby*-related release deficits. Our new variance-mean analysis (Fig. 7) demonstrates that both the reduction in N (release site number) and P_{rel} (release probability) correspond to the observed release deficits. As the reviewer will certainly be aware, even small deficits in Ca^{2+} currents driven by action potentials can have a significant impact on release probability due to the highly non-linear relationship between Ca^{2+} levels at AZs and neurotransmitter release rates. This reflects the high-order power relationship between Ca^{2+} concentration and Synaptotagmin function. That said, we acknowledge that a clear causal link remains to be definitively established.

6. *Since most of the discussion speculates on a potential role of Blobby in liquid-liquid phase separation (a hot and fashionable topic), this should be experimentally tested or other alternative explanations (eg direct interaction with other scaffold proteins) should be also discussed.*

The reviewer is absolutely correct in this regard, a point that has also been echoed by other reviewers. While we believe that the nature of *Blobby* suggests a potential role in liquid-liquid phase separation, we agree that it is too early to draw definitive conclusions at this point. In light of this, we have reduced the emphasis on this hypothesis in the discussion.

Additionally, we have incorporated the possibility that *Blobby* may directly interact with other scaffold proteins, a highly plausible alternative. It is tempting to speculate that *Blobby* binding might be needed to render BRP/RIM-BP into a "consumable state," possibly by exposing critical interaction surfaces needed for their binding to early scaffolding components. We now write in lines 332-335: "It is tempting to speculate, but certainly has to await appropriate experimental proof, that at nascent AZs in *Drosophila*, the IDRs of *Blobby* might play a role in

maintaining BRP/ELKS-containing condensates in a liquid state during a crucial step of the AZ assembly process.”

We also now write in the lines 386-388: “Future analyses should focus on determining which domains of Blobby, and its potential AZ interaction partners, are responsible for the observed deficits in SV release probability.”

Importantly both mechanisms could be involved in its function. By expanding the discussion in this way, we aim to provide a more balanced view of the potential molecular interactions in which Blobby may participate.

Minor comments:

We are extremely grateful to the reviewer for the effort in identifying these mistakes. All these mistakes have been corrected accordingly.

1. page and line numbers are helpful for reviewers.

Apologies for the earlier omission. We have now added page and line numbers to the manuscript and have referenced these when citing our edits and changes to ensure clarity and alignment with the reviewer's comments.

2. Merge the last two paragraphs in page 3 and delete the word however.

Has been corrected accordingly.

3. The sentence “the Blobby signal at presynaptic AZs is obviously of motoneuron means presynaptic origin”; seems incorrect.

Has been changed to: “Thus, Blobby protein at presynaptic NMJ AZs evidently is derived from the motoneurons, the presynaptic cell of the NMJ.” (Line 75-76)

4. Error in the legend of Fig. 2 E, F: RIM-BP instead of BRP.

5. Delete the word however at the end of page 4 after notably.

6. on page 6, paragraph 3, delete the word however.

Changes to points 4-6 have been made accordingly.

7. *Avoid describing HPF-EM technicalities and advantages in the main text (page 6).*

Good point. Has been reduced to the absolute minimum.

8. *Incorrect figures referred to on numerous occasions in the text (no figure number in the beginning of page 6, figure 4 instead of figure 5 later in page 6…). These are avoidable issues…*

This is a fully justified criticism, and we sincerely apologize for these avoidable errors. We have now made a thorough and substantial effort to correct all figure references and all other formal issues throughout the text to ensure accuracy and consistency. We are confident that such issues have been addressed fully.

Reviewer #5 (Remarks to the Author):

In this manuscript the authors identify and characterise a new protein that interacts with AZ proteins to facilitate SV trafficking. To identify this previously unannotated protein, the authors performed a Co-IP LC-MS experiment. The details surrounding these methods are very brief in the manuscript and require further explanation. Below are some specific comments:

Point-to-point response:

- 1. No details were provided in the methods section regarding the BRP IP, only references to published literature. It is okay to lean on published data to support the MM section, but a description of the work at a basic level is still necessary. At present the method cannot be evaluated or reproduced from this manuscript.*

This is a highly justified point. In response, we have now provided a more detailed description of the BRP immunoprecipitation (BRP-IP) procedure, which can be found in the Methods section. We hope it now provides enough detail, and reads as follows:

“The co-immunoprecipitation (Co-IP) experiment for Bruchpilot (BRP) was performed using crude synaptosomes resuspended in homogenization buffer (320 mM sucrose, 4 mM HEPES, and a protease inhibitor cocktail, pH 7.2), as described in ³. Approximately 6000 fly heads were collected per replicate, and synaptosomes were purified via differential centrifugation (see ³). For each replicate, 20 µg of gpBRPlast200 antibody was coupled to 50 µl of Protein A-coated agarose beads. A negative control was prepared by coupling gp-IgGs to beads. To prevent nonspecific binding of proteins to the beads, the synaptosome suspension was precleared by rotating for 1 hour at 4°C with naked beads. Following this, the bead-antibody and bead-IgG complexes were incubated overnight at 4°C with solubilized and precleared synaptosome membrane preparations (P2). After incubation, four washing steps (20 minutes each) were performed using immunoprecipitation (IP) buffer (containing 20 mM HEPES, pH 7.4, 200 mM NaCl, 2 mM MgCl₂, 1 mM EGTA, and 1% Triton X-100). Antibody-antigen complexes were eluted with 60 µl of 2x denaturing protein sample buffer and subjected to mass spectrometry and Western blot analysis.”

- 2. For the LC-MS analysis, a gel was run and in-gel tryptic digest performed. There is no information on the bands that were excised. Was it a brief gel run only for clean up? Or were proteins separated and digested in independent slices? There is insufficient information provided to understand or repeat the experiment.*
and

3. *There is insufficient information provided for the MS acquisition and analysis. What MS instrument was used to collect the data? What was the mass tolerance used during database searching? What modifications were set in the database searching?*

We agree with points 2 and 3 of the reviewer that additional details on sample processing and MS acquisition are necessary to ensure reproducibility. Accordingly, we have revised the relevant paragraphs in the methods section to include this information (see "Isolation and purification of *Drosophila* synaptosomes", "BRP co-immunoprecipitation", "Proteolytic digestion of BRP-IP eluate" and "LC-MS/MS analysis" in the Methods section.

4. *There is insufficient information provided for the MS acquisition and analysis. What MS instrument was used to collect the data? What was the mass tolerance used during database searching? What modifications were set in the database searching?*

Please refer to Response 2.

5. *In the volcano plot analysis and S_0 value and FDR cut off are described, but these criteria are not applied in the data visualization. Here only a non-adjusted P value is used, displaying no FDR correction. Additionally, the volcano plot appears truncated on the x-axis. It is better to show the complete data profile in the plot.*

We appreciate the reviewer bringing this error regarding the p-value to our attention. We have corrected the methods section, removing the claim of having used S_0 and an FDR cutoff. Additionally, we have included the full volcano plot as a new Reviewer Figure 8, while keeping the truncated version in the main manuscript to highlight proteins successfully enriched with the BRP antibody. Our intention was to label several proteins without overcrowding the visualization. We are prepared to include this information into the rebuttal if considered appropriate.

Reviewer Figure 8: Identification of Blobby as a co-precipitant from the Bruchpilot Immunoprecipitation.

A) The complete volcano plot comparing BRP immunoprecipitates (right side) with IgG control (left side) done in quadruplicate. X-axis: \log_2 fold-change value, Y-axis: $-\log_{10}$ of the p-value. **B)** BRP-IP eluate and IgG eluate (4 biological replicates each) were reduced (5 mM DTT at 37°C for 60 min), alkylated (40 mM CAA at RT for 30 min, dark) and loaded on NuPAGE 4 – 12% Bis-Tris SDS-PAGE. Subsequently, each lane was divided in three separate slices and independently subjected to an in-gel trypsin digestion procedure (enzyme:protein ratio of 1:20 (wt/wt) at 37°C overnight). Each slice was submitted for mass spectrometry analysis, resulting in 24 acquisitions and merged during data analysis.

6. *In the provided raw data table, I was unable to locate CG42795 in any searchable column (FASTA head or gene name). What is the associated protein identifier or gene name for this protein? The supporting data should be easily interpretable for the reader to look at. Additionally, I could not find the cited Unc-13A in the MS results, but rather Unc-13 as a master protein. There did not appear to be any evidence supporting the A isoform as the predominant protein detected.*

We appreciate the reviewer's observation regarding the missing gene name. This issue arises from the software MaxQuant, which does not always report all gene names. However, the gene name can be easily retrieved by searching the UniProt ID on the UniProt database. To facilitate the reader's access to this particular gene, we have now manually added the highlighted gene name (in the Volcano plot, Fig. 1A) to Supplemental table I. While we fully agree with the reviewer on the importance of well-organized supplementary data, we would also like to emphasize that we have uploaded the standard, unmodified MaxQuant output. This output provides a valuable resource beyond just the gene of interest, including important data quality indicators for each identified protein.

References:

1. Ramesh, N. *et al.* An antagonism between Spinophilin and Syd-1 operates upstream of memory-promoting presynaptic long-term plasticity. *Elife* **12** (2023).
2. Turrel, O., Ramesh, N., Escher, M.J.F., Pooryasin, A. & Sigrist, S.J. Transient active zone remodeling in the *Drosophila* mushroom body supports memory. *Curr Biol* **32**, 4900-4913 e4904 (2022).
3. Depner, H., Lutzkendorf, J., Babkir, H.A., Sigrist, S.J. & Holt, M.G. Differential centrifugation-based biochemical fractionation of the *Drosophila* adult CNS. *Nat Protoc* **9**, 2796-2808 (2014).